# CODE2BENCH: SCALING SOURCE AND RIGOR FOR DYNAMIC BENCHMARK CONSTRUCTION

**Zhe Zhang[1], Runlin Liu[1], Aishan Liu[1]\*, Xingyu Liu[1], Xiang Gao[1,2]\*, Hailong Sun[1,2]\***
[1]SKLCCSE, Beihang University
[2]Hangzhou Innovation Institute of Beihang University
{zhangzhe2023, runlin22, liuaishan}@buaa.edu.cn
{lxingyu, xiang_gao, sunhl}@buaa.edu.cn

## ABSTRACT

The evaluation of code-generating Large Language Models (LLMs) is fundamentally constrained by two intertwined challenges: a reliance on static, easily contaminated problem sources and the use of superficial, low-rigor testing. This paper introduces a new benchmark construction philosophy, **Dual Scaling**, designed to systematically address both limitations. Our approach involves continuously **scaling the source** of problems from dynamic, real-world code repositories and systematically **scaling the rigor** of tests via automated, high-coverage Property-Based Testing (PBT). We instantiate this philosophy in CODE2BENCH, an end-to-end framework that leverages Scope Graph analysis for principled dependency classification and a 100% branch coverage quality gate to ensure test suite integrity. Using this framework, we construct CODE2BENCH-2509, a new benchmark suite with native instances in both Python and Java. Our extensive evaluation of 10 state-of-the-art LLMs on CODE2BENCH-2509, powered by a novel "diagnostic fingerprint" visualization, yields three key insights: **(1)** models exhibit a fundamental performance gap, excelling at API application (Weakly Self-Contained tasks) but struggling with algorithmic synthesis (Self-Contained tasks); **(2)** a model's performance is profoundly shaped by the target language's ecosystem, a nuance we are the first to systematically quantify; and **(3)** our rigorous, scaled testing is critical in uncovering an "illusion of correctness" prevalent in simpler benchmarks. Our work presents a robust, scalable, and diagnostic paradigm for the next generation of LLM evaluation in software engineering. The code, data, and results are available at `https://code2bench.github.io/`.

## 1 INTRODUCTION

As Large Language Models (LLMs) are increasingly integrated into software development workflows Jimenez et al. (2023); Git; cur, the need for accurate and realistic evaluation of their coding capabilities has become paramount. However, the current landscape of code benchmarks is fundamentally constrained by two intertwined challenges: a reliance on **static, easily contaminated problem sources** and the use of **superficial, low-rigor testing**.

First, ❶ the *static* nature of canonical benchmarks like HumanEval Chen et al. (2021) and MBPP Austin et al. (2021) leads to an inevitable obsolescence; their problems, having existed for years, are likely part of LLM training corpora, turning evaluation into an exercise in memorization rather than true generalization Carlini et al. (2021); Sainz et al. (2023). While dynamic "live" benchmarks Jain et al. (2024a); Li et al. (2024b) have emerged, they often source problems from competitive programming, which may not reflect the complexity of real-world software engineering. Second, ❷ the *superficial testing* common to most benchmarks, often relying on a handful of example-based tests, creates an illusion of correctness. As highlighted by EvalPlus Liu et al. (2023a), this insufficient test rigor fundamentally limits their ability to uncover the subtle, edge-case failures that define the gap between functional code and production-ready software. As summarized in Table 1, existing methods fall short across key dimensions, highlighting the significant limitations remaining and the

---

*Corresponding authors.

Table 1: Comparison of CODE2BENCH-2509 with existing code generation benchmarks.

| Benchmark | Source | Dynamic | Deps Handled | Rigorous Test | Multi-lang Design |
|---|---|---|---|---|---|
| HumanEval Chen et al. (2021) | Manual | ✗ | ✗ | ✗ | ✗ |
| MBPP Austin et al. (2021) | Manual | ✗ | ✗ | ✗ | ✗ |
| EvalPlus Liu et al. (2023a) | Manual | ✗ | ✗ | ✓ | ✗ |
| LiveCodeBench Jain et al. (2024a) | Contests | ✓ | ✗ | ✓ | ✓ |
| RepoBench Liu et al. (2023b) | Project Codebases | ✗ | ✓ | ✗ | ✗ |
| HumanEval-X Zheng et al. (2023b) | Manual | ✗ | ✗ | ✗ | ✓ |
| BigCodeBench Zhuo et al. (2024) | Synthetic | ✗ | ✓ | ✗ | ✗ |
| DevEval Li et al. (2024c) | Project Codebases | ✗ | ✓ | ✗ | ✗ |
| EvoCodeBench Li et al. (2024b) | Project Codebases | ✓ | ✓ | ✗ | ✗ |
| **CODE2BENCH-2509 (Ours)** | **Project Codebases** | ✓ | ✓ | ✓ | ✓ |

necessity to design new benchmarks. To break this cycle of obsolescence and superficiality, we argue that a paradigm shift is needed. We propose a new benchmark construction philosophy centered on two core principles: **(1) Scaling the Source**, by dynamically and continuously ingesting a diverse array of problems from the ever-evolving landscape of real-world code repositories; and **(2) Scaling the Rigor**, by systematically generating comprehensive test suites with deep, verifiable coverage through Property-Based Testing (PBT) Claessen & Hughes (2000).

We instantiate this philosophy in **CODE2BENCH**, a novel, end-to-end framework that automates this dual-scaling process. ❶ To *Scale the Source*, CODE2BENCH first addresses the challenge of classifying diverse, real-world code. Our analysis of the existing benchmark landscape reveals an implicit bifurcation in evaluation focus, which we formalize into two primary task categories: **(1) Self-Contained (SC)** tasks, which require pure, dependency-free logic, reflecting the focus of benchmarks like HumanEval Chen et al. (2021) on core *algorithmic reasoning*; and **(2) Weakly Self-Contained (WSC)** tasks, which require the correct application of common libraries, capturing the focus of benchmarks like BigCodeBench Zhuo et al. (2024) on practical *API application*. This principled classification, enabled by our Scope Graph-based analysis, allows us to systematically generate tasks that target these distinct developer skills. ❷ To *Scale the Rigor*, CODE2BENCH then employs a powerful Property-Based Testing (PBT) engine and a stringent **100% branch coverage quality gate**. This ensures that every problem in our benchmark is not only realistic but also a fully-explorable logical challenge, backed by a test suite capable of deep, diagnostic validation.

Using this framework, we construct **CODE2BENCH-2509**, a new, multi-faceted benchmark suite with native instances in Python and Java, curated from recent, real-world repositories. Our extensive evaluation of 10 state-of-the-art LLMs on this suite demonstrates the power of our approach. The synergy of a scaled source and scaled testing enables an unprecedentedly fine-grained diagnostic, revealing: (1) a performance gap between models' ability in algorithmic synthesis (SC) and API application (WSC); (2) the profound impact of language paradigms on model failure modes, a nuance we are the first to systematically quantify; and (3) the critical role of rigorous testing in uncovering the "illusion of correctness" prevalent in simpler benchmarks.

We summarize our **contributions** as follows:

- We propose **CODE2BENCH**, a novel framework that introduces and operationalizes the **Dual Scaling** philosophy for benchmark construction, systematically scaling the source of problems with dynamic acquisition and the rigor of tests with a 100% coverage PBT quality gate.

- We construct and release **CODE2BENCH-2509**, a high-quality, contamination-resistant benchmark with native tasks in Python and Java, demonstrating significantly higher complexity and test rigor than prior work (Table 1).

- We provide a deep, **diagnostic analysis** of state-of-the-art LLMs, introducing novel visualizations and uncovering key insights into their strengths and weaknesses in real-world coding scenarios.

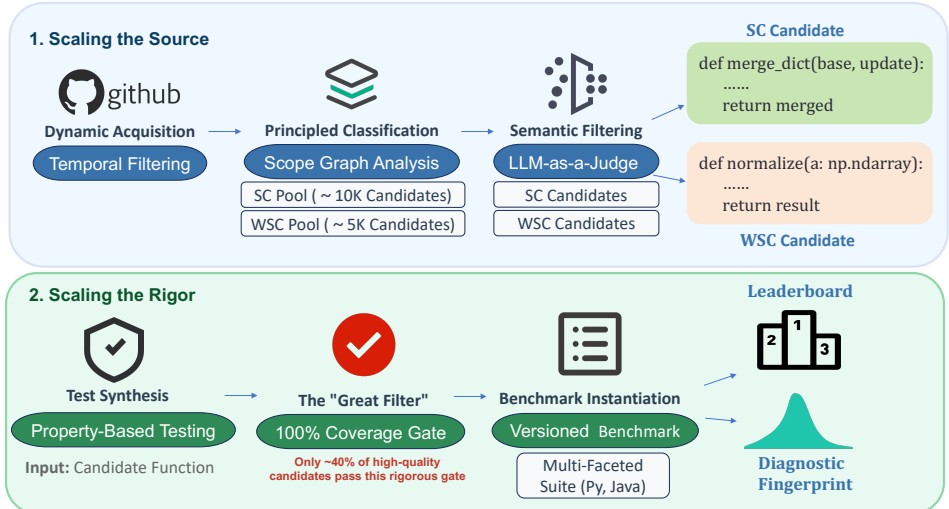

Figure 1: Overview of the CODE2BENCH Framework.

## 2 THE CODE2BENCH FRAMEWORK: A DUAL SCALING APPROACH

In this section, we detail the architecture and technical components of the CODE2BENCH framework. Our methodology is built upon the core philosophy of Dual Scaling: continuously **scaling the source** of benchmark problems to ensure realism and novelty, and systematically **scaling the rigor of our tests** to enable deep, diagnostic evaluation. We organize our discussion around these two principles.

### 2.1 SCALING THE SOURCE: DYNAMIC ACQUISITION FROM REAL-WORLD CODE

**Temporal Filtering for Contamination Resistance.** The primary threat to the validity of LLM evaluation is data contamination, where benchmarks become obsolete as their contents are absorbed into training corpora. To combat this inevitable obsolescence, our framework's first principle is to dynamically source problems that are provably unseen. We achieve this through **Temporal Filtering**, a deterministic strategy grounded in the version control timestamps of real-world code. Our method leverages a simple axiom: a model cannot have been trained on code that did not exist prior to its knowledge cutoff date. For each model under evaluation, we run our acquisition pipeline to extract functions exclusively from GitHub commits created after its official knowledge cutoff.

**Scope Graph-based Analysis for Dependency Classification.** To impose a meaningful structure on our diverse pool of functions, we automate their classification based on dependencies. We employ a **Scope Graph-based analysis** Néron et al. (2015)—a formal, language-agnostic method that precisely identifies all external dependencies, a task where simpler methods like AST traversal often fail.

Our classification algorithm is a deterministic, two-step process: (1) Dependency Identification, where we use the Scope Graph to compute the set of all *unresolved references* ($\mathcal{D}$) for each function. (2) Rule-based Classification, where we apply rules based on a predefined allowed libraries, $\mathcal{L}_{\text{allowed}}$:

- If $\mathcal{D} = \emptyset$, it is classified as **Self-Contained (SC)** (pure algorithmic reasoning).
- If $\mathcal{D}$'s dependencies resolve entirely within $\mathcal{L}_{\text{allowed}}$, it is **Weakly Self-Contained (WSC)** (API application).
- Otherwise, it is discarded (e.g., Project-Dependent).

This principled, automated process is a cornerstone of our framework, enabling the targeted evaluation of distinct model capabilities.

**Program Analysis for Testability and Complexity.** Following dependency classification, we apply a final layer of automated program analysis to ensure candidates are both testable and non-trivial.

First, to guarantee **testability**, we use Control-Flow Graph (CFG) analysis to discard functions lacking a verifiable, input-dependent output (e.g., no `return` statement). Second, to ensure a meaningful **complexity**, we filter functions based on their Cyclomatic Complexity McCabe (1976), targeting a range (e.g., $[2, 10]$) that balances challenge with solvability. This dual-filtering stage is crucial for curating a high-quality benchmark of tasks that are both evaluable and diagnostically valuable.

**LLM-based Semantic Filtering.** While prior analyses ensure structural soundness, they cannot distinguish a meaningful task from a trivial one. To assess for **semantic relevance** and **conceptual challenge**, we therefore employ an LLM-as-a-Judge Zheng et al. (2023a); Li et al. (2024a). This final filtering step, designed for high reliability via deterministic decoding and a structured classification prompt, ensures our benchmark contains problems of genuine substance. A rigorous validation, detailed in Appendix D, confirms its near-perfect agreement with human experts (Cohen's $\kappa = 0.95$).

## 2.2 SCALING THE RIGOR: AUTOMATED SYNTHESIS VIA PROPERTY-BASED TESTING(PBT)

**Property-Based Testing (PBT) for Comprehensive Input Generation.** Traditional example-based tests verify a function against a small, fixed set of known inputs. In contrast, Property-Based Testing (PBT) Claessen & Hughes (2000) explores a much larger input space by generating hundreds or thousands of random, yet structured, inputs and asserting that a general *property* of the code holds true for all of them. In our framework, the core property we test is **functional equivalence** with the ground-truth implementation. For any valid input $\mathbf{x}$ generated by our PBT engine, the output of an LLM-generated function $f_{\text{LLM}}(\mathbf{x})$ must match the output of the original, real-world ground-truth function $f_{\text{gt}}(\mathbf{x})$. The ground-truth function thus serves as a perfect **test oracle**.

The process of input synthesis is driven by automated **strategy generation**. For each function candidate, our framework analyzes its signature, including parameter types and type hints, to compose a set of PBT *strategies*. These strategies are not simple random generators, but intelligent explorers of the input domain, designed to produce a rich distribution of values—including typical inputs, boundary cases (e.g., empty lists, zeros, min/max values), and complex nested structures (e.g., variable-shaped lists of dictionaries). This automated process yields a comprehensive suite of hundreds of input-output pairs $(\mathbf{x}_i, f_{\text{gt}}(\mathbf{x}_i))$ for each function, forming the foundation for the rigorous validation described next. The specific PBT libraries used for each language (e.g., Hypothesis for Python, jqwik for Java) are detailed in Appendix E.

**The "Great Filter": A 100% Coverage Quality Gate.** While Property-Based Testing generates a high volume of diverse inputs, quantity alone does not guarantee rigor. A test suite, however large, is only effective if it thoroughly exercises the internal logic of the function under test. To enforce this level of rigor systematically, we introduce the final and most stringent stage of our pipeline: the **"Great Filter"**, a quality gate that mandates **100% branch coverage**.

The mechanism is as follows: after a PBT suite is synthesized for a ground-truth function $f_{\text{gt}}$, we execute the entire suite against $f_{\text{gt}}$ itself and measure the resulting branch coverage using standard language-specific tools (e.g., `coverage.py`). A function candidate and its corresponding test suite are only accepted into the final benchmark if and only if this execution achieves 100% branch coverage. This seemingly simple requirement has a profound impact on the final benchmark's quality and character. It acts as a powerful, dual-purpose filter:

- **It filters out inadequate tests.** If a PBT suite fails to achieve full coverage, it indicates that the input generation strategy was not sophisticated enough to explore all logical paths of the function. Such a test suite would be incapable of providing a truly rigorous evaluation, and is discarded.

- **It filters out untestable functions.** More importantly, if even a well-designed PBT strategy cannot trigger all branches, it often signals that the function itself is "untestable" in isolation. This typically occurs in functions with defensive code for unreachable states, complex error handling coupled to external systems, or other logic that cannot be exercised through its public API. These functions, while present in real-world code, are unsuitable for a standalone, functional correctness benchmark.

The "Great Filter" is the primary reason for the significant reduction in candidates observed in our data funnel (as shown in Figure 1). It is a deliberate trade-off, prioritizing **uncompromising rigor over sheer volume**. The result is a smaller, but significantly more potent, benchmark where every

single problem is guaranteed to be a non-trivial, fully-explorable logical puzzle, backed by a test suite capable of validating every branch of its solution.

**Instruction Generation for Task Specification.** To ensure a fair and effective evaluation, each task is accompanied by a clear, unambiguous instruction. Our framework automates the generation of these instructions by refining a function's original source docstring and signature using a powerful LLM (GPT-4o) with deterministic decoding. To mitigate potential biases, we also employ a back-translation perturbation technique Zhuo et al. (2024); Wang et al. (2022); Dhole et al. (2021).

Crucially, the instruction style is systematically adapted to the task's dependency classification, ensuring the model is provided with the precise context needed for the specific challenge:

- **For SC Tasks(e.g., SC-Python, SC-Java):** Instructions use language-native conventions—Python docstrings with types like `list` and `dict` for SC-Python, and Javadoc with Java types like `List<String>` for SC-Java. Though library-free, these tasks assess a model's proficiency with each language's core built-in features and data structures.
- **For WSC Tasks:** The instruction is made **library-aware** and **language-native**. It explicitly names the required external libraries (e.g., NumPy) and uses the precise, idiomatic types of the target language's ecosystem (e.g., `numpy.ndarray`). This targets the evaluation on a model's practical ability to correctly apply common APIs.

**Benchmark Instantiation and Runner Generation.** The final step of our framework packages each curated problem into an executable benchmark instance. This instance comprises two key components: a test suite and a test runner. The **test suite**, containing hundreds of input-output pairs, is generated using a **native Property-Based Testing (PBT) engine** (e.g., `Hypothesis`, `jqwik`) to ensure high coverage and rigor. To conduct the evaluation, a corresponding **language-native Test Runner** is automatically generated. The runner is responsible for deserializing the test suite, executing the LLM-generated code, and performing a rigorous deep comparison against the ground-truth outputs. To guarantee the runner's correctness, we perform a **dry run** where the LLM's function is replaced by the ground-truth function, ensuring the entire test harness passes flawlessly before evaluation. This end-to-end native approach, validated by a dry run, ensures our evaluation is both stringent and reliable. Further details are in Appendix F.

## 3 THE CODE2BENCH-2509 BENCHMARK SUITE

Table 2: Quantitative characteristics of the `CODE2BENCH-2509`, compared to prior benchmarks.

| Metric / Dimension | SC-Python | WSC-Python | SC-Java | *HumanEval* | *MBPP* |
|---|---|---|---|---|---|
| **I. Scale & Complexity** | | | | | |
| # Tasks | 217 | 194 | 249 | 164 | 974 |
| Avg. Lines of Code (LoC) | **20.6** | 18.3 | **14.1** | 7.3 | 6.5 |
| Avg. Cyclomatic Complexity (CC) | **5.3** | 2.6 | **3.6** | 2.8 | 2.3 |
| Difficulty (E:M:H Ratio) | 0.30:0.40:0.30 | 0.28:0.41:0.31 | 0.27:0.43:0.30 | - | - |
| **II. Testing Rigor** | | | | | |
| Avg. Test Cases per Task | **~500** | **~500** | **~500** | ~7.8 | ~3.0 |
| Test Coverage Guarantee | **100% Branch** | **100% Branch** | **100% Branch** | *Variable* | *Variable* |
| **III. Diversity & Extensibility** | | | | | |
| Source Type | Real-World | Real-World | Real-World | Hand-Crafted | Crowd-Sourced |
| Dependency Scope | Self-Contained | **>30 Libraries** | Self-Contained | Self-Contained | Self-Contained |
| Language Extensibility | (Python) | (Python) | **Java Native** | (Python) | (Python) |

CODE2BENCH-2509 is a new benchmark suite, automatically curated from May to September 2025, designed to overcome the limitations of prior benchmarks by systematically expanding evaluation along three key dimensions: **Testing Rigor**, **Dependency Level**, and **Framework Extensibility**. Figure 2 visually situates our benchmark within this landscape, showcasing its significant leap forward compared to predecessors like HumanEval and BigCodeBench. We targeted actively maintained, open-source repositories on GitHub. To minimize noise and bias, we enforced the following criteria: (1) **Community Validation:** Repositories must have ≥ 500 stars; (2) **Active Maintenance:** Commits within the last 3 months; and (3) **Domain Diversity:** We employed stratified sampling across 10 diverse domains (e.g., Web, ML, System) to prevent over-representation of any single field. We actively filtered out homework assignments and tutorials. A detailed disclosure of the selection

criteria, sampling strategy, and the full repository list is provided in Appendix H.

The quantitative evidence for this advancement is detailed in Table 2. Our instances demonstrate significantly higher structural complexity (e.g., an average Cyclomatic Complexity of **5.3** for `SC-Python` vs. 2.8 for HumanEval) and an order-of-magnitude increase in testing rigor, featuring **~500** test cases per task with a guaranteed **100% branch coverage**. Beyond these metrics, the suite's high quality is rooted in the rich diversity of its tasks, sourced from **220 Python and 189 Java** recent, real-world repositories. This ensures wide topical coverage, while the successful instantiation of a native Java suite provides concrete proof of our framework's extensibility. This combination of complexity, rigor, diversity, and extensibility provides a more challenging and realistic platform for assessing the true capabilities of modern LLMs. More details can be found in Appendix H.

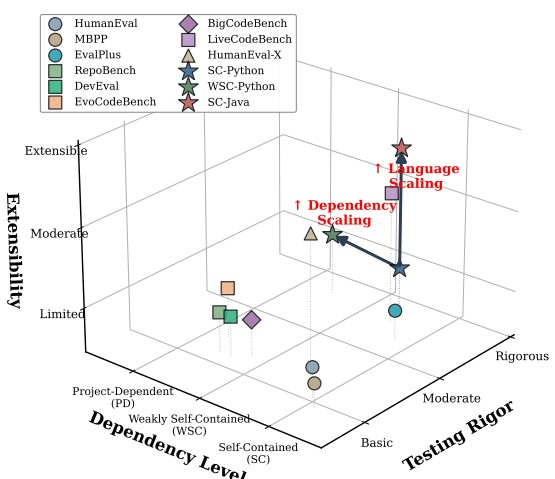

Figure 2: The multi-dimensional evaluation landscape.

## 4 EVALUATION

### 4.1 EXPERIMENTAL SETUP

**Evaluated Models.** We selected a diverse suite of 10 state-of-the-art Large Language Models, encompassing both leading closed-source APIs and prominent open-source families. A cornerstone of our evaluation integrity is the strict prevention of data contamination. The `CODE2BENCH-2509` benchmark was constructed exclusively from code committed after May 2025, a date subsequent to the knowledge cutoff of all evaluated models.

**Evaluation Protocol.** Our primary metric is **Pass@1** Kulal et al. (2019), which measures the functional correctness of the first-generated solution and closely mirrors a developer's real-world experience with coding assistants Jain et al. (2024a). All evaluations were conducted in a zero-shot, deterministic setting, employing greedy decoding (temperature 0) as is standard practice Zhuo et al. (2024); Roziere et al. (2023). For each task, the model received a standardized instruction containing the function signature and a natural language description(See more details in Appendix I.2).

**Execution Environment.** The generated code for each task was executed in a sandboxed environment against the full suite of PBT-generated tests (~500 per task). A language-specific test runner performed differential testing, comparing the output of the model-generated code against the ground-truth implementation. A task is considered passed only if it correctly solves all test cases. Open-source models were served via vLLM Kwon et al. (2023), while others were accessed through their official APIs.

### 4.2 A MULTI-DIMENSIONAL DIAGNOSTIC OF LLM CAPABILITIES

A primary limitation of existing benchmarks is their inability to provide deep, diagnostic insights. We argue this stems from two fundamental constraints: a narrow source of problems and superficial testing. The `CODE2BENCH` framework overcomes these limitations through two core principles: **scaling the source** from dynamic, real-world code, and **scaling the rigor** of evaluation via Property-Based Testing (PBT). In this section, we demonstrate how this dual-scaling approach enables an unprecedentedly fine-grained diagnostic of LLM coding capabilities.

Table 3 presents the overall Pass@1 performance, revealing clear capability tiers among models. The data indicates that performance varies significantly across our three benchmark components,

Table 3: Pass@1 performance (%) on the `CODE2BENCH-2509` suite.

| Model | SC-Python (%) [95% CI] | WSC-Python (%) [95% CI] | SC-Java (%) [95% CI] |
|---|---|---|---|
| *Closed-Source Models* | | | |
| Claude-4-sonnet | **40.1** [33.6 − 46.5] | 38.7 [32.0 − 45.4] | 47.4 [40.9 − 53.4] |
| Gemini-2.5-Flash | 37.8 [30.9 − 44.2] | 36.6 [29.4 − 43.3] | 45.0 [39.0 − 51.0] |
| *Open-Source Models (Ordered by Scale)* | | | |
| DeepSeek-V3 | 34.4 [28.4 − 40.4] | 37.6 [31.4 − 44.3] | **47.8** [41.4 − 54.2] |
| Qwen3-235b-a22b | 34.6 [28.6 − 41.0] | 36.6 [29.9 − 43.3] | 46.6 [40.9 − 53.0] |
| Llama-4-scout | 25.8 [19.8 − 31.8] | 32.5 [26.3 − 39.2] | 44.2 [37.8 − 49.8] |
| Qwen3-32b | 31.3 [25.8 − 36.9] | 34.5 [27.8 − 41.2] | 43.0 [37.4 − 49.4] |
| Mistral-small-3.1 (24B) | 30.4 [24.4 − 36.9] | **38.7** [32.5 − 45.9] | 43.4 [37.4 − 49.4] |
| Qwen3-8b | 25.1 [19.3 − 31.4] | 34.0 [27.8 − 40.7] | 39.0 [32.9 − 44.2] |
| Gemma-3n-e4b-it | 22.6 [17.1 − 28.6] | 26.3 [19.6 − 32.5] | 34.5 [28.5 − 40.2] |
| Qwen3-1.7b | 14.3 [9.7 − 19.4] | 16.5 [11.3 − 21.6] | 17.7 [12.8 − 22.5] |

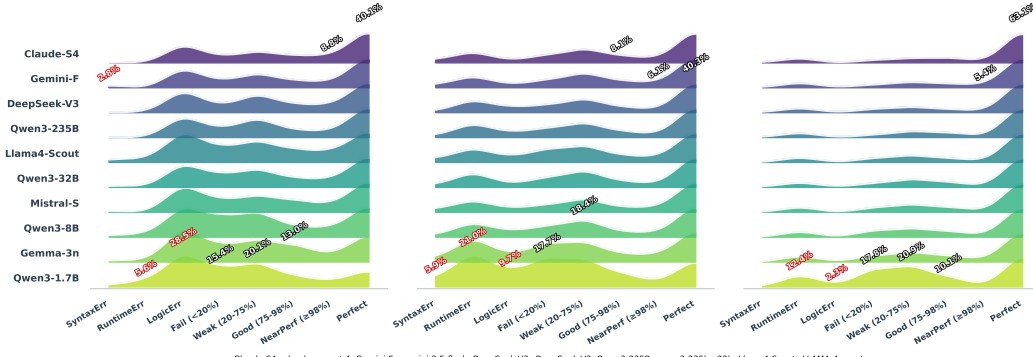

Claude-S4=claude-sonnet-4; Gemini-F=gemini-2.5-flash; DeepSeek-V3=DeepSeek-V3; Qwen3-235B=qwen3-235b-a22b; Llama4-Scout=LLAMA-4-scout
Qwen3-32B=qwen3-32b; Mistral-S=mistral-small-3.1-24b-instruct; Qwen3-8B=qwen3-8b; Gemma-3n=gemma-3n-e4b-it; Qwen3-1.7B=qwen3-1.7b

Figure 3: Fingerprints across the three evaluation tracks—`SC-Python` (left), `WSC-Python` (middle), and `SC-Java` (right)—shown as ridgeline plots. Each curve captures a model's outcome distribution, ranging from `SyntaxErr` to `Perfect`, with key pass rates annotated.

hinting at the different skills they evaluate. To move beyond these aggregate scores and understand the underlying reasons for these variations, we now turn to a more fine-grained analysis.

To dissect *how* and *why* models succeed or fail, we introduce a granular, multi-stage outcome spectrum, visualized as a diagnostic fingerprint for each model in Figure 3. Outcome categories are ordered to reflect a progression from catastrophic failure to complete success: **SyntaxErr** (code fails to compile/run), **RuntimeErr** (code crashes during execution), **LogicErr** (code runs but is logically incorrect on all test cases), followed by a four-tier partial success range based on pass rates—from **Fail (<20%)** to **NearPerf (>=98%)**—and culminating in **Perfect** solutions. Figure 3 visualizes each model's distribution across this spectrum. The synergy between the aggregate scores in the table and these distributional fingerprints reveals two profound insights into the nature of LLM coding intelligence.

**Decoupling Algorithmic Synthesis from API Application.** The diagnostic fingerprints (Figure 3, left vs. middle) show a systematic shift in failure modes: on `SC-Python`, the dominant failure is `LogicErr`, indicating a core challenge in first-principle reasoning. Conversely, on `WSC-Python`, this peak vanishes and `RuntimeErr` emerges as a primary obstacle, suggesting the challenge shifts to the correct application of external APIs.

**Language Paradigms as Performance Scaffolding.** The comparison between `SC-Python` and `SC-Java` (Figure 3, left vs. right), reveals the profound impact of language paradigms. In Java, the prominent `LogicErr` and `RuntimeErr` peaks seen in Python are sharply suppressed, while

performance in the `Perfect` category surges for all models. We hypothesize this is not because models are inherently "better" at Java, but because its static type system acts as a powerful **"performance scaffolding"**, pruning a vast space of potential errors at compile time. This demonstrates that an LLM's coding ability is not an abstract quantity but is fundamentally intertwined with the target language's ecosystem, a crucial interaction our framework is the first to systematically quantify.

### 4.3 THE EFFECTIVENESS OF PBT-GENERATED TESTS

A core tenet of the CODE2BENCH framework is **scaling test rigor** via Property-Based Testing (PBT). This section provides quantitative evidence for the necessity of this approach by analyzing the prevalence of "Near-Perfect" failures—solutions that pass at least 98% of our comprehensive test suite but fail on a handful of subtle edge cases. These instances represent an "illusion of correctness" that would likely go undetected by conventional, less rigorous benchmarks.

Table 4 quantifies the frequency of these "near-miss" failures, where a solution passes the vast majority of test cases (>98%) but ultimately fails. The data reveals a significant and consistent pattern: **on average, 6.94% of submissions for `SC-Python` tasks fall into this treacherous category.**

This finding directly underscores the critical necessity of our Property-Based Testing (PBT) methodology. Without the exhaustive, edge-case-driven verification enabled by PBT, these nearly 7% of submissions—which are functionally almost correct—would have been falsely classified as successes by conventional, sparse test suites. This would lead to a significant overestimation of model capabilities.

Top-performing models are not immune to this illusion of correctness; "DeepSeek-V3" and "Claude-4-sonnet", for example, see approximately 8% of their submissions fall into this category. This demonstrates that even the most capable models consistently struggle with the final frontier of logical robustness, a weakness that only a truly rigorous testing paradigm like PBT can reliably expose.

Interestingly, the rate of near-perfect failures is lower in `WSC-Python` (4.57%) and lowest in `SC-Java` (2.25%). This aligns with our findings in Section 4.2. In WSC tasks, the problem is often a binary choice of the correct API call, leaving less room for "almost correct" logic. In Java, the strict type system likely prevents many of these subtle logical errors at the compi-

Table 4: Prevalence of "Near-Perfect" Failures (Pass@ ≥98%) in CODE2BENCH.

| Model | SC-Py (%) | WSC-Py (%) | SC-Java (%) |
|---|---|---|---|
| Claude-Sonnet-4 | **8.76** | 5.15 | 2.41 |
| Gemini-2.5-Flash | 6.45 | **5.67** | **4.02** |
| DeepSeek-V3 | 7.80 | 3.61 | 2.41 |
| Qwen3-235b-a22b | 6.45 | 3.61 | 2.01 |
| LLAMA-4-scout | 8.29 | 5.15 | 2.01 |
| Mistral-Small-3.1 | 6.91 | **5.67** | 2.01 |
| Qwen3-32b | 7.37 | 3.61 | 1.61 |
| Qwen3-8b | 6.76 | 4.64 | 2.41 |
| Gemma-3n-e4b-it | 5.53 | **5.67** | 2.81 |
| Qwen3-1.7b | 5.07 | 3.09 | 0.80 |
| **Avg. (%)** | **6.94** | **4.57** | **2.25** |

lation stage. Therefore, the high rate of near-perfect failures in `SC-Python` highlights its unique position as the most challenging testbed for a model's pure, unaided logical robustness.

Ultimately, this analysis validates the central role of rigorous, scaled testing. By systematically uncovering these near-perfect failures, CODE2BENCH provides a more accurate measure of a model's true capabilities and offers invaluable, fine-grained feedback for identifying and rectifying their most subtle weaknesses.

### 4.4 THE IMPACT OF DYNAMIC SOURCING AND REAL-WORLD COMPLEXITY

To situate CODE2BENCH-2509 within the existing landscape, we conduct a direct comparison against EvalPlus, a state-of-the-art benchmark that enhances HumanEval/MBPP with more rigorous, mutation-based testing. While EvalPlus represents the pinnacle of evaluation on static, well-known problem sets, CODE2BENCH-2509 introduces the dimensions of **dynamic sourcing** and **real-world complexity**. This comparison aims to answer a critical question: how does a model's performance on canonical programming puzzles translate to its ability to handle fresh, complex code from the wild?

The head-to-head comparison is visualized in Figure 4, which plots the Pass@1 scores of ten prominent LLMs on our `SC-Python-2509` (X-axis) against their performance on HumanEval (Y-axis). The results are stark and reveal three critical insights:

**A Systematically More Challenging Benchmark.** The most striking observation is that all models are located deep in the **red-shaded region**, far above the $y = x$ diagonal of equal performance. This demonstrates that CODE2BENCH-2509 presents a systematically higher level of difficulty for all models, without exception. For instance, top-performing models like Claude-4-Sonnet, which achieve a near-perfect score of 97% on HumanEval, see their performance plummet to **40.1%** on our benchmark—a drop of over 50 percentage points. This substantial performance gap suggests that high scores on legacy benchmarks may create an illusion of capability that does not hold up against the complexity of novel, real-world code.

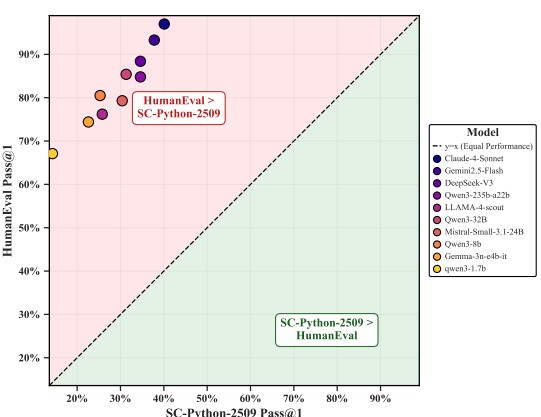

**Probing Generalization Over Memorization.** This performance delta is not merely a matter of difficulty, but of the fundamental capabilities being measured. HumanEval's problems are years old and likely part of the

Figure 4: Performance on Evalplus and CODE2BENCH-2509

training corpora. The lower performance on `SC-Python-2509`—a benchmark guaranteed to be unseen—strongly indicates that our evaluation measures a model's true generalization ability on novel algorithmic challenges, rather than its capacity for pattern memorization. This underscores the critical need for dynamic, contamination-resistant benchmarks to ensure a fair and realistic assessment of an LLM's problem-solving intelligence.

## 5 RELATED WORK

**Code Generation Benchmarks** Evaluating Large Language Models (LLMs) on code generation tasks is an active research area, with numerous benchmarks proposed. Early benchmarksCassano et al. (2023); Li et al. (2022) like HumanEval Chen et al. (2021), MBPP Austin et al. (2021), and APPS Hendrycks et al. (2021) provide static collections of isolated code snippets with tests, proving valuable for initial model development but facing limitations regarding dataset contamination Carlini et al. (2021); Sainz et al. (2023); Yang et al. (2023); Team et al. (2024) and the lack of real-world context and dependencies. Efforts like EvalPlus Liu et al. (2023a) enhance static benchmarks with more robust tests via mutation. However, these may not fully represent real-world code complexities or provide automated construction from code repositories. Multilingual benchmarksYan et al. (2023) like HumanEval-X Zheng et al. (2023b), Aider polyglot benchmark pol and AutoCodeBench Chou et al. (2025) evaluate cross-language abilities but are at risk of data leakage. Benchmarks focusing on repository-level context or dependenciesTang et al. (2023); Li et al. (2024b); Yu et al. (2024); Wang et al. (2024) include RepoBench Liu et al. (2023b), CrossCodeEval Ding et al. (2023), R2E Jain et al. (2024b), DevEval Li et al. (2024c), BigCodeBench Zhuo et al. (2024), and CODEAGENT Zhang et al. (2024). WebBench Xu et al. (2025) introduces sequential, real-world web development tasks. While these capture aspects of real-world interaction, they often lack sufficient testing. CODE2BENCH stands out by offering an end-to-end pipeline for dynamically generating rigorous benchmark instances from recent real-world GitHub repositories.

**Data Leakage and Live Benchmarks** A key limitation of static benchmarks is the risk of test set contamination, where models may have been trained on the same data used for evaluationCarlini et al. (2021); Sainz et al. (2023); Yang et al. (2023); Team et al. (2024). This has motivated the development of "live" benchmarks. DynaBench Kiela et al. (2021) identified these challenges and advocated for continuously evolving benchmarks. Chatbot Arena Chiang et al. (2024) provides a platform for dynamic evaluation based on user interactions. LiveBench White et al. (2024) sources new data from specific, non-code domains like mathematics and news, while LiveCodeBench Jain et al. (2024a) collects recent problems from competitive programming platforms. Other methods leverage LLMs

for task mutation to generate new problems, such as EvoEval Xia et al. (2024). Evocodebench Li et al. (2024b) introduces a periodically updated benchmark to mitigate leakage, while DOMAINEVAL Zhu et al. (2025) employs dynamic data sources with automated updates for the same purpose. Both efforts currently focus on Python and may lack rigorous test. Arena-Hard-Auto Li et al. (2024d) filters crowdsourced prompts into benchmarks. While existing initiatives have made progress in addressing contamination and enabling dynamic evaluation, they often rely on specific data sources, focus on evaluation platforms, or lack a systematic, automated pipeline for constructing high-quality benchmarks from real-world code at scale. CODE2BENCH bridges this gap by providing a novel, automated, end-to-end pipeline that dynamically extracts, filters, and constructs rigorous benchmark.

## 6 CONCLUSION & FUTURE WORK

We introduced **Dual Scaling**, a new philosophy for benchmark construction, and presented **CODE2BENCH**, a framework that operationalizes it by systematically scaling the source of problems from real-world code and the rigor of tests via high-coverage PBT. Our evaluation on the resulting `CODE2BENCH-2509` benchmark provided a deep, diagnostic analysis of modern code LLMs, revealing a consistent gap between their API application (WSC) and algorithmic synthesis (SC) capabilities, and quantifying for the first time how language paradigms shape their failure modes.

**Limitations and Future Work.** Our work, while establishing a robust framework for evaluating functional correctness, opens several avenues for future expansion. We plan to extend our **Scaling the Source** principle to repository-level, Project-Dependent (PD) tasks to assess codebase understanding. Concurrently, we will expand our **Scaling the Rigor** principle to incorporate non-functional properties such as code efficiency and security. By evolving along these axes, CODE2BENCH will continue to provide a challenging and realistic measure of true software engineering competence.

## ACKNOWLEDGEMENT

This work was partly supported by Natural National Science Foundation of China under Grant No. 62472017, and partly by Guangxi Collaborative Innovation Center of Multi-source Information Integration and Intelligent Processing.

## REPRODUCIBILITY STATEMENT

We are committed to ensuring the full reproducibility of our work. The complete CODE2BENCH-2509 benchmark suite, including all task instructions, ground-truth solutions, and PBT-generated test suites scripts, is also included. The project, including all data and results, is also available at our anonymized repository: `https://code2bench.github.io/`.

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

## A PREPROCESSING AND DATA STRUCTURING

This appendix details the preprocessing and data structuring steps performed in the Function Filtering pipeline (Section 2) to transform raw source code into a standardized representation suitable for subsequent analysis.

### A.1 SOURCE CODE PARSING VIA ABSTRACT SYNTAX TREES (ASTS)

Following the initial identification of candidate functions from the source code repositories, the raw text code of these functions and their surrounding context is processed using Tree-sitter tree sitter (2024). Tree-sitter is a parser generator tool that produces concrete syntax tree parsers for various programming languages. Unlike traditional compilers that focus on semantic analysis, Tree-sitter is specifically designed for source code analysis tools, providing robust, incremental parsing capabilities and generating detailed, well-structured Concrete Syntax Trees (CSTs), which are closely related to Abstract Syntax Trees (ASTs).

For each identified function, the corresponding source code snippet is parsed into its AST representation. The AST is a tree structure that represents the abstract syntactic structure of source code written in a programming language. Each node in the tree denotes a construct occurring in the source code (e.g., function definition, variable declaration, expression, statement). Parsing the raw code into an AST is crucial as it:

- Standardizes the code representation, abstracting away syntactic variations (e.g., whitespace, comments) and providing a consistent structure regardless of the original formatting.
- Exposes the hierarchical and relational structure of the code, making it amenable to systematic program analysis techniques.

### A.2 EXTRACTION OF RELATIONAL INFORMATION

From the generated ASTs, we extract essential relational information and metadata for each candidate function. This extracted information is crucial for the subsequent dependency analysis, program analysis, semantic filtering, and benchmark instance construction stages. Key information extracted includes:

- Function Signature: The function name, parameters, and their declared types or type hints (if available). This information is directly used for generating the benchmark instruction.
- Source Code Snippet: The exact lines of code corresponding to the function definition, serving as the Ground Truth implementation and the basis for program analysis.
- Import Statements: Identification of modules or names imported within the function's scope or in its surrounding file. This is vital for understanding potential external dependencies.
- Metadata: Information such as the original file path and commit hash, aiding in traceability and ensuring the recency of the code source.

This structured extraction process transforms the raw, unstructured code into a queryable and analyzable format, laying the foundation for the automated benchmark construction pipeline.

## B  SCOPE GRAPH BASED DEPENDENCY ANALYSIS

This appendix provides a formalized technical explanation of the Scope Graph-based dependency analysis employed in the Function Filtering stage of the CODE2BENCH. This analysis is fundamental to identifying and controlling external dependencies of candidate functions, enabling the classification of tasks into Self-Contained (SC) and Weakly Self-Contained (WSC) categories crucial for rigorous and standardized evaluation.

### B.1  SCOPE GRAPH MODEL

After parsing the source code of a project into Abstract Syntax Trees (ASTs) using Tree-sitter, we construct a Scope Graph $G = (V, E)$. The vertex set $V$ includes:

- Scope nodes $S \subseteq V$, representing hierarchical scopes like modules, classes, functions, or blocks. Each scope $s \in S$ represents a region of code where identifiers are defined and resolved.

- Definition nodes $D \subseteq V$, representing the points where identifiers are defined (e.g., variable declarations, function definitions). Each definition $d \in D$ corresponds to a specific identifier name $id(d)$.

- Reference nodes $R \subseteq V$, representing the points where identifiers are used (referenced) within the code. Each reference $r \in R$ corresponds to a specific identifier name $id(r)$.

The edge set $E$ includes:

- Scope hierarchy edges $E_{scope} \subseteq S \times S$, representing nested scopes (e.g., a function scope nested within a class scope).

- Definition edges $E_{def} \subseteq S \times D$, representing that a definition $d$ is contained within a scope $s$.

- Reference edges $E_{ref} \subseteq S \times R$, representing that a reference $r$ occurs within a scope $s$.

- Binding edges $E_{bind} \subseteq R \times D$, representing that a reference $r$ resolves to a definition $d$ according to the language's scoping rules. These edges are established during the resolution process.

Thus, $V = S \cup D \cup R$ and $E = E_{scope} \cup E_{def} \cup E_{ref} \cup E_{bind}$.

### B.2  DEPENDENCY RESOLUTION PROCESS

For each function candidate $F$, we identify the set of all identifier references $R_F \subseteq R$ occurring within its body scope $s_F$. For each reference $r \in R_F$, the dependency resolution process attempts to find a corresponding definition $d \in D$ such that a binding edge $(r, d) \in E_{bind}$ can be established by traversing the Scope Graph $G$ outwards from $s_F$ according to the language's lexical scoping rules.

A reference $r \in R_F$ is classified as an **unresolved reference** if no definition $d \in D$ is found within the analysis scope of $G$ that $r$ can legally bind to. Let $U_F \subseteq R_F$ be the set of all unresolved references for function $F$. These unresolved references $U_F$ represent the external dependencies of function $F$.

The Scope Graph approach offers conceptual language-agnosticism as the graph structure and resolution mechanism are based on universal programming concepts (scopes, bindings), abstracted from specific syntax by the AST input from Tree-sitter. Language-specific scoping rules are encoded in how the resolution traversal and binding edges $E_{bind}$ are determined.

### B.3  SC/WSC CLASSIFICATION BASED ON DEPENDENCIES

Based on the set of unresolved references $U_F$ and the function's import statements, we classify function $F$:

Let $L_{allowed}$ be the predefined set of allowed common external libraries. For each unresolved reference $r \in U_F$, we attempt to determine its origin based on the function's import statements and knowledge of library APIs. A reference $r$ is considered resolved to an allowed library $l \in L_{allowed}$ if its name $id(r)$ corresponds to an identifier provided by library $l$, and library $l$ is correctly imported or accessible within the scope of function $F$.

Let $U_{F,allowed} \subseteq U_F$ be the subset of unresolved references that resolve to identifiers within libraries in $L_{allowed}$.

- **Self-Contained (SC)**: Function $F$ is classified as SC if and only if its set of unresolved references is empty, i.e., $U_F = \emptyset$. This means all identifiers are defined locally or are language built-ins.
- **Weakly Self-Contained (WSC)**: Function $F$ is classified as WSC if $U_F \neq \emptyset$ and all unresolved references resolve to allowed libraries, i.e., $U_F = U_{F,allowed}$.
- **Discarded**: Function $F$ is discarded if $U_F \neq \emptyset$ and $U_{F,allowed} \subsetneq U_F$. This means there are unresolved references that are not from allowed libraries.

This systematic, formalized approach, leveraging the Scope Graph representation, provides a precise and robust method for controlling dependencies and classifying functions, which is essential for the reliability and scalability of the CODE2BENCH.

# C  PROGRAM ANALYSIS FOR TESTABILITY AND COMPLEXITY

This appendix provides further details on the program analysis techniques employed in the Function Filtering stage (Section 2) to ensure candidate functions are functionally testable and represent non-trivial coding challenges. Our analysis focuses on properties derived from the Control Flow Graph (CFG) and the structural complexity of the function implementation.

## C.1  CONTROL FLOW ANALYSIS FOR TESTABILITY

To identify functions amenable to automated testing and output verification, we perform Control Flow Graph (CFG) analysis. For a given candidate function $f$, its CFG represents all possible execution paths through the function's code. Nodes in the CFG correspond to basic blocks of code (sequences of instructions executed sequentially), and directed edges represent potential control transfers between these blocks (e.g., branches, loops, function calls).

We analyze the structure of the CFG to filter out functions that inherently lack verifiable output. Specifically, we identify functions where:

- The CFG contains no paths leading to a `return` statement. Such functions typically perform actions (e.g., printing to console, modifying global state) without providing a value that can be easily captured and compared against an expected output in an automated differential testing setup.
- All paths leading to a `return` statement return only constant values, or values derived solely from constants without any dependency on input parameters or complex intermediate computations. While these functions have a return value, their behavior is trivial and does not require probing with diverse inputs.

Functions matching these criteria are excluded from the candidate pool, as they are either difficult to test functionally in isolation or do not represent meaningful code generation tasks for evaluating LLM capabilities beyond simple retrieval.

## C.2  COMPLEXITY ASSESSMENT VIA CYCLOMATIC COMPLEXITY

To focus the benchmark on tasks that require models to generate non-trivial code logic, we assess the structural complexity of each candidate function using Cyclomatic Complexity ($CC$) Cyc. Cyclomatic Complexity is a quantitative measure of the number of linearly independent paths through a program's source code. It is calculated based on the CFG of the function using the formula:

$$CC = E - N + 2P$$

where:

- $E$ is the number of edges in the CFG.
- $N$ is the number of nodes in the CFG.
- $P$ is the number of connected components in the CFG (for a single function, $P = 1$).

Therefore, for a single function, $CC = E - N + 2$. $CC$ is strongly correlated with the number of decision points (e.g., `if`, `while`, `for`, `case`) in the code, providing an estimation of the code's logical complexity.

We employ $CC$ as a filter to select functions that fall within a desirable complexity range, avoiding tasks that are either too simple to be challenging or excessively complex to be solvable or reliably testable as isolated benchmark instances. Specifically, we define a range $[CC_{min}, CC_{max}]$ (e.g., $[2, 10]$) and include only functions whose calculated $CC$ falls within this range.

- Functions with $CC < CC_{min}$ (e.g., $CC < 2$) typically represent very simple linear code or trivial control flow structures that offer little challenge.
- Functions with $CC > CC_{max}$ (e.g., $CC > 10$) may indicate highly complex logic, potentially involving deep nesting, numerous branches, or intricate loops, which can be challenging for LLMs to generate correctly and for automated tests to cover exhaustively. Furthermore, such high complexity in a real-world function might often be coupled with complex, uncontrolled dependencies.

# D    SEMANTIC FILTERING AND DIFFICULTY ASSESSMENT

This appendix provides additional technical details regarding the LLM-based Semantic Filtering process used in the Candidate Filtering stage to refine the candidate function set through semantic assessment. While the main text outlines the motivation and overall structure, here we present in detail the prompts employed during this stage of semantic filtering.

To validate the robustness and objectivity of our LLM-based filtering, we conducted two inter-rater reliability studies. **Inter-LLM Agreement.** We measured the agreement between three diverse, state-of-the-art LLM judges (GPT-4o, Claude-4-Sonnet, and a Qwen3-Max) on a random sample of 100 function candidates. For the binary task of semantic filtering (trivial vs. meaningful), the judges achieved a Fleiss' Kappa of 0.92, indicating almost perfect agreement. **Human-LLM Agreement.** We further compared the judgments of our primary LLM judge (GPT-4o) against a consensus gold standard established by two human experts with extensive programming experience. On a set of 50 functions, the analysis yielded a Cohen's Kappa of 0.95.

Table 5: Prompt Template for Self-Contained Ground-Truth Filter in CODE2BENCH

---

# System
## Task Description
You are an expert in the field of coding, tasked with determining whether a given Python function is suitable for generating an instruction (question).
The function will be analyzed based on its characteristics, functionality, and adherence to specific criteria. If the function meets the criteria, it is deemed suitable; otherwise, it is not.
## Criteria for Suitability To determine whether a function is suitable for generating an instruction, consider the following criteria:
### 1. Function Parameters
- **Basic Types Only**: The function's parameters must be basic types (e.g., 'int', 'float', 'str', 'list', 'dict', etc.)...
### 2. Function Complexity
- **Meaningful Complexity**: The function should provide a meaningful test of the model's capabilities...
### 3. Side Effects and Dependencies
- **No Side Effects**: The function should not have side effects (e.g., modifying global variables, writing to files, etc.).
- **No External Imports**: The function should not import other modules or depend on external libraries...
## Output Format If the function is **suitable**, return:
```json
{
"Suitable": true,
"Reason": "The function meets all criteria for generating an instruction."
}
```
If the function is not suitable , return:

...
## Examples

...
## Note
- Ensure that your analysis is thorough and considers all aspects of the function.
- Provide clear and concise reasoning for your decision. - Only return the Json.

---

# User
Please check the last result:
**[Last Result]**
Error response:
**[Error Response]**
**[Function Message]**

---

Table 6: Prompt Template for Weakly Self-Contained Ground-Truth Filter in CODE2BENCH

---

# System
You are an expert in Python coding, tasked with determining whether a given Python function meets the requirements below for generating a benchmark. The function is weakly self-contained, meaning it depends only on Python standard libraries or specific external libraries (e.g., numpy, re, pandas) and no custom modules. You will analyze the function based on its characteristics, functionality, and adherence to specific criteria. If the function meets the criteria, it is deemed suitable; otherwise, it is not.

## Criteria for Suitability
To determine whether a function is suitable, consider the following criteria:

### 1. Function Parameters
- **Basic and Library Types**: Parameters must be basic Python types (e.g., 'int', 'float', 'str', 'list', 'dict') or types from standard/external libraries (e.g., 'numpy.ndarray', 're.Pattern').
- If the parameters' type is missing, but you can infer it from the code, it is **suitable**.
- If the function relies on methods or attributes of unknown objects, it is **not suitable**. But if the function uses lib types (e.g., 'numpy.ndarray', 'pandas.DataFrame'), it is **suitable**.

### 2. Function Complexity
- **Meaningful Complexity**: The function should provide a meaningful test of the model's capabilities, with clear logic and purpose.
- If the function is overly long but trivial (e.g., repetitive assignments), it is **not suitable**.
- If the function is too simple (e.g., basic getter/setter), it is **not suitable**.

### 3. Domain Knowledge
- **General Applicability**: The function should not require highly specialized domain knowledge to understand or implement...

### 4. Property-Based Testing
- **Constructible Inputs**: The function should allow generating random inputs for property-based testing to verify its behavior.

## Output Format
If the function is **suitable**, return:
```json
{
"Suitable": true,
"Reason": "The reason why the function meets all criteria for generating a benchmark."
}
```
If the function is not suitable, return:
```json
{
"Suitable": false,
"Reason": "The reason why the function is not suitable for generating a benchmark."
}
```

## Examples
...

## Note
- Provide clear and concise reasoning for the decision.
- If the function meets our standards but is missing imports, it is still suitable.
- If the function only uses 'typing' for type hints, it is not suitable.
- Only return the JSON output in the specified format.

---

# User
Please check the last result:
**[Last Result]**
Error response:
**[Error Response]**
**[Function Message]**

Table 7: Prompt Template for Weakly Self-Contained Difficulty Assessment in CODE2BENCH

---

# System
## Task Description
You are an expert code analyst tasked with assessing the difficulty level of a weakly self-contained Python function. A weakly self-contained function depends only on Python standard libraries or specific external libraries (e.g., NumPy, pandas, re) and no custom modules. Assume the function is valid and suitable for analysis. Assign a difficulty level of "Easy", "Medium", or "Hard" based on the complexity of its logic, structure, required concepts, and cognitive load to understand.
## Criteria for Difficulty Assessment
### Easy Difficulty - **Logic**: Very simple, minimal or no branching, single loop or direct parameter use.
- **Structure**: Short, linear, immediately clear control flow.
- **Concepts**: Basic Python constructs (variables, operators, lists, strings) or simple library calls (e.g., Counter from collections, basic re matching, pandas filtering).
- **Cognitive Load**: Minimal; purpose and execution are obvious at a glance.
- **Example**:
```python
from collections import Counter
def count_word_frequencies(text: str) -> dict[str, int]:
"""Count the frequency of each word in a text string.
Args:
text: Input string containing words.
Returns:
Dictionary mapping words to their frequency.
"""
words = text.lower().split()
return dict(Counter(words))
```

...
### Medium Difficulty - Logic: Moderate complexity, with loops, conditions, or data transformations (e.g., - filtering, sorting, deduplication).
- Structure: Traceable control flow, possibly nested loops or multiple steps, moderate length.
- Example:
...
Hard Difficulty
- Logic: Complex, with nested loops, intricate transformations, non-trivial algorithms (e.g., multi-dim aggregation, complex grouping), or subtle edge cases.
- Structure: Dense or multi-step control flow, significant state management.
Example:
...
## Output Format Return ONLY a JSON object containing the assessed difficulty level:
```json
{
"Difficulty": "Easy/Medium/Hard"
}
```
## Note
- weakly self-contained function depends only on Python standard libraries or specific external libraries (e.g., NumPy, pandas, re) and no custom modules.
- Focus on logic and structure, not the library's complexity (e.g., simple Counter usage is Easy, complex NumPy array ops are Hard).
- Analyze the function's code, docstring, and logic thoroughly.
- Only return the JSON output in the specified format.

---

# User
[Function Message]

---

Table 8: Prompt Template for Self-Contained Difficulty Assessment in CODE2BENCH

# System
## Task Description
You are an expert code analyst. Your task is to assess the difficulty level of the provided Python function. Assume the function is generally valid and suitable for analysis. Assign a difficulty level of "Easy", "Medium", or "Hard" based on the complexity of its logic, structure, required concepts, and cognitive load to understand.
## Criteria for Difficulty Assessment
### Easy Difficulty
- Logic: Straightforward, minimal branching (simple if/else), possibly a single simple loop. Direct use of parameters.
- Structure: Typically short, linear control flow. Easy to follow step-by-step.
- Concepts: Relies on fundamental programming constructs (variables, basic operators, standard data types, simple function calls).
- Cognitive Load: Low; the function's purpose and execution are immediately apparent.
- Example:
```python
def parse_message(message: str) -> str:
if message is None:
return ""
message = message.strip().lower()
# Simple string checks and manipulations
if not message.startswith(("run-slow", "run_slow", "run slow")):
return ""
message = message[len("run slow") :]
while message.strip().startswith(":"):
message = message.strip()[1:]
return message
```

...
### Hard Difficulty
- Logic: Moderate complexity. May involve nested loops, multiple non-trivial conditions, manipulation of data structures (e.g., iterating through lists/dicts with transformations), implementing a common simple algorithm, or tracking state across iterations.
- Structure: Control flow is more involved but still reasonably traceable. Function length might be moderate.
- Example:

...
### Hard Difficulty
- Logic: Complex logic. Might involve recursion, implementing non-trivial algorithms.
- Structure: Can have nested structures, complex control flow, significant state management, or rely on clever interactions between code parts. May not be long but could be dense.
- Example:

...
## Output Format
Return ONLY a JSON object containing the assessed difficulty level:
{
"Difficulty": "Easy/Medium/Hard"
}

# User
[Function Message]

# E    PROPERTY-BASED TESTING

This appendix provides additional technical details regarding the Property-Based Testing (PBT) process to generate rigorous test cases for CODE2BENCH. While it describes the core principles, here we elaborate on the technical implementation.

Our PBT approach is centered around defining strategies for generating diverse, valid inputs for a given function and verifying a core property against the ground truth implementation.

## E.1    STRATEGY BUILDING AND INPUT SYNTHESIS

The Strategy Builder component analyzes the function's signature, parameter types, and inferred constraints from type hints, docstrings, and static analysis of the ground truth code. It then leverages a PBT library (e.g., Hypothesis for Python) to compose generation **strategies** for each function parameter. These strategies are designed to explore a wide range of valid inputs, including typical values, edge cases (e.g., empty lists, zero, maximum/minimum values), and combinations of different input types within complex structures (e.g., lists of dictionaries, tuples of specific types). The process is constraint-aware, ensuring generated inputs adhere to inferred conditions.

For example, for a function taking a list of integers 'def process(data: list[int])', the strategy might generate lists of varying lengths, containing both positive and negative integers, zeros, and potentially boundary values like 'sys.maxint'. For a function taking a string with specific format requirements, the strategy would be built to generate strings adhering to that format.

## E.2    PROPERTY DEFINITION AND VERIFICATION

The core **property** we verify for generated code is **functional equivalence** with the ground truth implementation. For every input $x_i$ generated by the strategies, we compute the expected output $y_i = f_{GT}(x_i)$, where $f_{GT}$ is the ground truth function. The PBT framework then requires that for any generated input $x_i$, the output of the generated code $f_{LLM}(x_i)$ must equal $y_i$. Any input $x_i$ where $f_{LLM}(x_i) \neq y_i$ constitutes a test case failure, and the PBT framework can then attempt to "shrink" $x_i$ to find a minimal failing input.

## E.3    ENSURING TEST RIGOR AND COVERAGE

To ensure the generated test suites are truly rigorous, we incorporate a quality control step based on test coverage. After generating a suite of $(x_i, y_i)$ pairs using PBT strategies, we execute these test cases against the ground truth implementation itself. We use code coverage tools (e.g., "coverage.py" for Python) to measure the branch coverage achieved by the generated test suite on the ground truth code. Only test suites that achieve a high coverage threshold (e.g., 100% average branch coverage) are accepted and included in the final benchmark instance. This filtering step is crucial: even if a strategy can generate many inputs, if those inputs don't exercise the complex branching logic of the function, the resulting test suite is not rigorous enough to effectively verify model implementations.

## E.4    CODE EXAMPLE (PYTHON/HYPOTHESIS)

Below is a simplified illustrative example using the Python Hypothesis library to demonstrate how strategies and properties are defined.

```python
import hypothesis.strategies as st
from hypothesis import given, settings, Verbosity

# 1. Define strategies for input generation
# Strategy for generating arbitrary strings
string_strategy = st.text()

def ground_truth_reverse(s: str) -> str:
    return s[::-1]
```

```python
def llm_generated_reverse(s: str) -> str:
    # Hypothetical LLM output, might have bugs
    return "".join(reversed(s))

# Define a property using @given decorator
@given(s=string_strategy) # Use the defined strategy to generate input
↪    's'
@settings(max_examples=1000) # Example settings for testing
def test_reverse_property(s):
    # Property: Reversing twice returns the original string
    assert reverse_string(reverse_string(s)) == s # This property tests
    ↪    the function against itself

    # Property: LLM output matches Ground Truth output
    expected_output = ground_truth_reverse(s)
    actual_output = llm_generated_reverse(s)
    assert actual_output == expected_output # This property tests LLM
    ↪    output against GT
```

This example illustrates the basic concept of defining strategies for input generation and asserting properties about the function's behavior for these inputs. In the CODE2BENCH pipeline, these principles are automated and scaled to generate comprehensive test suites for a wide variety of functions extracted from real-world code.

## F    TESTCASE RUNNER GENERATION

This appendix provides a detailed presentation of the prompts used for generating test runners (Section 2). Since test runners are inherently language-specific, we designed tailored prompts for each programming language. The corresponding prompts are listed in Tables 9 through 10.

Table 9: Prompt Template for SC-Java Testcase Runner Generation in CODE2BENCH

---

**# System**
**## Task Description**
As an expert Java developer specializing in test case generation and function signature translation, your task is to generate a Java test file and function signature based on a Python function and its test cases. The test file will load test cases from a JSON file and execute tests using a provided 'Helper.deepCompare' method.
**### Requirements**
1. **Java Test File**:
- Generate a complete, executable Java test file in the 'p0' package.
...
2. **Function Signature**:
- Provide only the function signature in the 'p0.Tested' class.
...
3. **Special Considerations**:
- **Type Safety**:
- Ensure 'TestCase' fields exactly match JSON keys and types (e.g., 'lines' → 'List<String>', 'line_index' → 'int').
...
4. **Type Definition Rules**
- Follow these rules to determine where to define types:
| Usage Scenario | Location | Example
| |————————-|—————|————————|
| Used in function signature | 'tested.java' | 'public static class TagInfo {}' |
| Used in both | 'tested.java' | Shared types always in implementation |
**## Input Format**
- **Test Cases JSON**: A JSON array of test cases, provided as '{testcases_str}'.
- **Python Function**: A Python function to be tested, including its signature and implementation, provided as code.
**## Output Format**
'''plaintext

**[Java test file]**

<signature>
**[Java function signature]**
</signature>
**## Examples**
...
**## Note**
- Ensure the generated Java test file is complete and executable.
- The function signature should be a valid Java function signature that matches the Python function's behavior.
- Only return the Java test file in '' and the function signature in '<signature>' tags.

---

**# User**
The previously generated runner code:
**[Runner Code]**
The previously generated runner code resulted in the following error during execution:
**[Error Message]**
**[Function Message]**

---

Table 10: Prompt Template for WSC-Python Testcase Runner Generation in CODE2BENCH

---

# System
## Task Description
You are an expert Python developer specializing in property-based testing and test case execution. Your task is to generate a **complete and executable Python script** that loads test cases from a JSON file generated by a Hypothesis-based Testcase Generator and re-runs the test logic to verify the behavior of a function under test ('func1') against a ground truth function ('func0'). The functions depend only on standard libraries or specific external libraries (e.g., NumPy, re) and no other custom modules. The script will:

1. Load test cases from the JSON file ('test_cases.json') containing 500 test cases, each with a '"Inputs"' dictionary mapping 'func0''s argument names to JSON-serializable values.
2. Re-run the test logic by calling 'func0' (ground truth) and 'func1' (under test) with the loaded inputs, comparing their outputs via differential testing.
3. Compare outputs using:
- For basic types and their combinations ('int', 'float', 'str', 'list', 'dict', etc.), use 'deep_compare' from the 'helper' module.
- For third-party library types (e.g., 'numpy.ndarray', 'tuple' of 'numpy.ndarray'), use library-provided comparison functions (e.g., 'np.allclose') or custom logic if none is provided.
4. Report test results, indicating whether each test case passes or fails, with detailed failure information (inputs, expected output, actual output).
## Input The input is a Python Testcase Generator script that includes:
1. The ground truth function 'func0', its dependencies (e.g., 'numpy', 're'), and implementation.
2. Hypothesis strategies and '@example' decorators defining input generation logic.
3. A test function ('test_<function_name>') that generates and saves 500 test cases to 'test_cases.json', each containing only '"Inputs"'.
Provided in '<Testcase Generator>' tags.
## Output Generate a complete, executable Python script that:

...
## Example

...
## Note - Focus on Loading and Re-testing: Load test cases from test_cases.json and verify func1 against func0 using differential testing.
- Preserve Input Format: Ensure inputs match func0's signature, converting JSON-serialized inputs (e.g., list to np.ndarray) as needed.
- Output Comparison:
- Use deep_compare from helper for basic types and combinations (int, float, str, list, dict, etc.).
- Use library-provided comparisons (e.g., np.allclose for numpy.ndarray) for third-party library types, or custom logic if none is provided.
- Executable Code: The script must be complete, self-contained, and executable.
- Differential Testing: Since test cases contain only inputs, compute expected outputs by calling func0 and compare with func1's outputs.
- External Libraries: Include imports for func0's dependencies (e.g., numpy, re) and handle their data types (e.g., np.ndarray).

---

# User
The previously generated runner code:
**[Runner Code]**
The previously generated runner code resulted in the following error during execution:
**[Error Message]**
**[Function Message]**

---

# G INSTRUCTION GENERATION

Table 11: Prompt Template for WSC-Python Instruction Generation in CODE2BENCH

---

**# System**

You are a python programming expert who is refining docstrings in existing programs. You will be given a python function in a python file with an existing (possibly underspecified) docstring with corresponding some input-output examples extracted.

Your goal is to refine the associated docstring by making it more informative, precise and complete without adding verbosity or detailed programming logic to the docstring. When there is a docstring, the docstring is used to evaluate the code generation capabilities of a model.

The docstring should particularly describe the format and types of the expected inputs and output as well as the behavior of the function. Do not guess outputs for functions. Finally, do not throw away existing details from the docstrings and only insert content you are sure about. Do NOT have repeated content in the docstring and ONLY describe the high-level function behavior without going into implementation details.

**## Requirements**

1. **Core Description Fidelity**: The docstring must accurately reflect the function's behavior and describes the task this function solves. **Pay close attention to the sequence of checks, conditions, and resulting actions within the code.

2. ** Highlight any **special rules** that affect the model's correct understanding of the function's behavior, such as:

- Recursive behavior
- Merging, flattening, filtering, transformation logic
- Edge cases or type-specific handling
- Magic numbers or constants

3. **Docstring Refinement**: If the function already has a docstring, integrate and refine its content to meet these requirements. Do not discard existing accurate information.

4. **Conditional Example Handling**:

You should judge whether to include examples based on the original docstring's content:

- **If the original docstring contains an 'Examples' section**:
- Preserve all original examples **verbatim** in the final docstring's 'Examples' section.
- Format them clearly in Language-Agnostic(e.g., showing input and expected output).
- Do **not** add or modify examples from the 'Example Usages' data.

**## Input** "'python
{ground_truth_function_code}
"'

**### Output Format**

- Return the docstring in <docstring> tags following the Google-style format.
- Include the function signature in <signature> tags, with a TODO placeholder for the implementation.

**#### Example output:**

...

**## Note**

- Only add examples in Docstring when the function already has an 'Examples' section in the docstring. Do not add examples from the 'Example Usages' data if the original docstring does not contain 'Examples'.

- If the signature has type hints, import nessesary types from the standard library (e.g., 'from typing import List, Dict') in signature.

---

**# User**
<function>
[Function Code]
</function>
<example usages>
[Example Usage]
</example usages>

Table 12: Prompt Template for SC-Python Instruction Generation in CODE2BENCH

---

# System
You are a programming documentation architect specializing in creating precise, implementation-agnostic specifications. Generate a docstring that enables accurate reimplementation in any programming language. When there is a docstring, the docstring is used to evaluate the code generation capabilities of a model.

## Requirements
1. **Core Description Fidelity**: The docstring must accurately reflect the function's behavior and describes the task this function solves. **Pay close attention to the sequence of checks, conditions, and resulting actions within the code.

2. Highlight any **special rules** that affect the model's correct understanding of the function's behavior, such as:
- Recursive behavior
- Edge cases or type-specific handling
- Magic numbers or constants
- Special settings that may affect the difficulty for others to correctly implement functions based on docstrings. e.g., the Ground Truth function may add some special string at the end of the result, so the docstring should mention this case, otherwise, the model may not be able to implement the function correctly.

2. **Language-Agnostic Terminology**: Use universal concepts for types and logic.
- Describe parameters and return values using **conceptual types** (e.g., "an integer", "a boolean value", "a sequence of numbers", "a text string") instead of language-specific type hints ('int', 'bool', 'list', 'str').
- Describe operations conceptually (e.g., "checks if X contains Y", "iterates over the elements", "applies a function to each element") rather than Python built-ins ('s1.find("'")', 's1.replace').

3. **Docstring Refinement**: If the function already has a docstring, integrate and refine its content to meet these requirements. Do not discard existing accurate information.

4. **Conditional Example Handling**:
- **If the original docstring contains an 'Examples' section**:
- Preserve all original examples **verbatim** in the final docstring's 'Examples' section...

## **Input Structure**
'''python
{ground_truth_function_code}
'''

### Example Usages:
{example_usage_data}
(Note: Example Usages will be provided in a format like input/output pairs.)

## **Output:**
Only return the docstring content in <docstring> tags and the function signature in the <signature> tags. The docstring should be enclosed in triple double quotes ('"""Docstring goes here"""'). The function signature should be formatted in "'python' code block with the function name and a TODO comment indicating where the implementation should go.

## Note:
- Only add examples in Docstring when the function already has an 'Examples' section in the docstring. Do not add examples from the 'Example Usages' data if the original docstring does not contain 'Examples'.
- If the signature has type hints, import nessesary types from the standard library (e.g., 'from typing import List, Dict') in signature.

---

# User
<function>
[Function Code]
</function>
<example usages>
[Example Usage]
</example usages>

---

Table 13: Prompt Template for SC-Java Instruction Generation in CODE2BENCH

# System
You are an expert Java architect and technical writer, specializing in creating high-quality, professional Javadoc documentation. Your task is to generate a precise and informative Javadoc for a given Java method, enabling another senior Java developer to re-implement it accurately within a complete, self-contained 'Tested.java' file.

## Core Objective
The generated Javadoc and method signature must serve as a perfect "specification" for the provided ground-truth method.

## Requirements
1. Core Description Fidelity: The Javadoc must accurately reflect the method's behavior, including the precise sequence of checks, conditions, and resulting actions within the code.
2. Edge Cases: Detail how the method handles edge cases, such as null, empty, or special/magic values.
3. Data Structures: Accurately describe parameters and return values using standard Java types.
4. Javadoc Refinement: If the method already has a Javadoc, your primary goal is to refine and enhance it to meet these high standards. Integrate existing accurate information with your new insights. Do not discard valuable details from the original author.
5. Example Handling:
* If the original Javadoc contains an example (e.g., in a '<pre>@code ...</pre>' block): Preserve the original example verbatim.
* If not: Omit any example section entirely.

## **Input Structure**
'''java
{ground_truth_java_method_code}
'''

## Output Structure
Return a single '<signature>' tag containing a complete, self-contained, and runnable 'Tested.java' file content. The content must be enclosed in a '''java' code block and include all necessary imports, the generated Javadoc, the 'public class Tested', and the public static method signature with a '// TODO: implement this method' comment.
<signature>
'''java
// All necessary imports (e.g., java.util.*) should be here.
import java.util.List;
import java.util.Map;
public class Tested {
/**
* High-quality, Google-style Javadoc goes here. ...
*/
public static ReturnType methodName(ParameterType parameterName) {
// TODO: implement this method
}
}
'''
</signature>

## Final Instructions
- Ensure the method signature in the '<signature>' tag perfectly matches the ground truth, including visibility ('public static'), return type, method name, and parameter types.
- Don't add implementation details.
- Include all necessary imports based on the types used in the signature.

# User
<function>
[Function Code]
</function>

# H    BENCHMARK

## H.1    BENCHMARK DETAILS

This appendix provides a detailed breakdown of the construction process and diversity analysis for the `CODE2BENCH-2509` suite. All data was sourced from public GitHub repositories with commits made between May 2025 and September 2025.

To ensure the diversity, quality, and representativeness of the source code used in CODE2BENCH, we adhered to a strict, multi-stage data selection protocol. This appendix details the criteria and sampling strategies employed during the "Scaling the Source" phase.

We targeted high-quality, actively maintained, and well-tested open-source repositories hosted on GitHub. To be included in our initial candidate pool, a repository must meet the following quantitative thresholds:

- **Community Validation:** $\geq 500$ Stars. This threshold filters out personal experiments and ensures a baseline of community scrutiny and adoption.
- **Active Maintenance:** At least one commit within the 3 months prior to our data collection cutoff (May 2025). This ensures the code reflects modern coding practices and library versions.
- **Test Availability:** Must contain an identifiable test suite (e.g., presence of `tests/` directory, usage of `pytest`/`junit`). This is crucial for verifying our ground truth extraction.
- **License Permissibility:** Must be under permissive licenses (e.g., MIT, Apache 2.0, BSD) to allow for redistribution and modification.

To mitigate noise and bias, we applied semantic filters to exclude repositories that do not represent real-world software engineering contexts:

- **Homework & Tutorials:** Repositories with keywords like "assignment", "course", "tutorial", "learn-python/java", or "leetcode" in their description or README were excluded. These typically contain toy problems distinct from production-grade code.
- **Aggregators & Forks:** We excluded repositories identified as mere collections of other projects (e.g., "awesome-xxx") or direct forks without significant divergence, ensuring unique data sources.

To prevent domain bias (e.g., over-representation of web frameworks), we employed a stratified sampling strategy based on GitHub topics and repository tags. We categorized candidates into 10 primary domains covering the spectrum of software development:

1. **Web Development** (e.g., frameworks, clients)
2. **Data Science & Machine Learning** (e.g., analytics, pipelines)
3. **System Utilities** (e.g., cli tools, file manipulation)
4. **Network & Protocol** (e.g., async io, sockets)
5. **Security & Cryptography**
6. **Database & Storage**
7. **Text Processing & NLP**
8. **Media & Graphics** (e.g., image processing)
9. **Scientific Computing**
10. **Development Tools** (e.g., linters, parsers)

We performed random sampling within each stratum to build our final source list of Python and Java repositories.

To fully support reproducibility and auditability, the complete list of source repositories is available on our open-source project page: `https://code2bench.github.io/`.

## H.2 The Data Funnel: From Raw Functions to a Gold Standard

The final size of our benchmark is a direct result of a stringent, multi-stage filtering pipeline designed to prioritize quality, realism, and rigor. Table 14 illustrates this "Great Filter" process for the Python components, starting from over one million recently updated functions identified in 220 repositories. This rigorous process ensures that only the most suitable and high-quality candidates become part of the final benchmark.

Table 14: The Data Funnel for `CODE2BENCH-2509` Python components, illustrating the multi-stage filtering process that prioritizes quality and rigor.

| Stage | Filtering Action & Criteria | SC Candidates | WSC Candidates |
|---|---|---|---|
| 1. Initial Pool | Functions parsed from recent commits | ~1.17 Million | |
| 2. Dependency Filter | Scope Graph: Strictly SC / WSC compliant | 27,649 | 12,335 |
| 3. Testability/Complexity | Testable outputs & Cyclomatic Complexity [2,10] | 7,102 | 3,278 |
| 4. Sub-sampling | Breadth-first sampling for diversity | 901 | 432 |
| 5. Semantic Filter | LLM-as-a-judge removes trivial tasks | 479 | 315 |
| **6. The Great Filter** | **PBT: 100% Branch Coverage Guarantee** | **217** | **194** |

## H.3 Task Diversity Analysis

The high quality of `CODE2BENCH-2509` is rooted in its rich diversity of tasks and application domains.

### H.3.1 SC-Python: Algorithmic and Real-World Logic

The 217 tasks in the `SC-Python` component cover a wide spectrum of real-world programming challenges beyond simple puzzles. Key functional categories include:

- **Text Processing & Formatting**: Ranging from LaTeX sanitization (`strip_latex`) to complex wrapping for SVG elements (`wrap_text_for_svg`).

- **Classic & Modern Algorithms**: Includes fundamental algorithms like `levenshtein` and `binary_search`, as well as logic relevant to modern development tools like parsing LCOV reports (`parse_lcov`).

- **Parsing & Extraction**: Challenges models to parse structured data from unstructured text, such as extracting JSON from noisy strings (`_extract_balanced_json`) or parsing version numbers (`parse_version`).

- **AI/LLM-Specific Logic**: A unique feature is the inclusion of tasks from the AI ecosystem, such as formatting LLM error messages (`format_llm_error_message`) and parsing model outputs (`extract_weave_refs_from_value`).

- **Complex Data Structure Manipulation**: Requires deep understanding of nested structures (e.g., `flatten_state_dict`, `deep_merge`).

### H.3.2 WSC-Python: A Broad and Realistic API Ecosystem

The 194 tasks in the `WSC-Python` component require the use of over **35 distinct libraries and modules**[1], ensuring a faithful evaluation of a model's practical API fluency. The distribution is representative of real-world Python development, covering a wide and diverse ecosystem rather than focusing on a narrow set of APIs. The required libraries span multiple key domains of software engineering:

- **Data Processing & Text Manipulation**: A significant portion of tasks involve core data handling using standard libraries like re, `json`, `ast`, `datetime`, `urllib`, `unicodedata`, and `base64`.

---

[1]This count is based on unique top-level import statements, e.g., re, `numpy`, `scipy.spatial`.

- **Scientific Computing & Data Science**: The benchmark probes capabilities in specialized numerical and data-centric domains, requiring libraries such as `numpy`, `scipy` (including submodules like `scipy.spatial.transform`), and `pandas`.
- **Machine Learning**: Uniquely, our suite includes tasks that interact with the machine learning ecosystem, leveraging modules from `scikit-learn` like `TfidfVectorizer` and `cosine_similarity`.
- **Advanced Standard Library Proficiency**: Beyond common utilities, the tasks require a deep knowledge of Python's standard library, including advanced modules such as `itertools`, `collections` (e.g., `Counter`, `defaultdict`), `difflib`, `bisect`, and `struct`.

This broad and realistic library coverage ensures that `WSC-Python` provides a holistic assessment of a model's ability to function as a practical coding assistant in a diverse range of real-world scenarios.

### H.3.3 SC-JAVA: VALIDATING EXTENSIBILITY WITH DIVERSE TASKS

The successful generation of the 249-task `SC-Java` suite provides concrete evidence of our framework's extensibility. The quality of this component is validated by its high diversity, which mirrors the real-world complexity found in its Python counterpart. The tasks span a wide array of application domains:

- **String Manipulation & Parsing**: A large portion of tasks involve complex string operations, such as format conversion (`convertToCamelCase`), cleaning (`cleanText`), and escaping for different contexts (`escapeJsonString`, `escapeCsvField`).
- **Encoding & Data Conversion**: Numerous tasks focus on byte-level manipulation, primarily converting between byte arrays and hexadecimal strings (e.g., `bytesToHex`), a common task in systems programming and networking.
- **Mathematics & Algorithms**: The suite includes non-trivial algorithms like checksum validation (`luhnBankCardVerify`) and edit distance calculation (`levenshteinDistance`).
- **Domain-Specific Logic**: Crucially, the tasks are not generic puzzles but are rooted in specific application domains, including game development (e.g., Minecraft metadata transformation, `transformMetaDecoModel`) and systems utilities (e.g., calculating CPU affinity masks, `maskToCpuAffinity`).

This demonstrates our framework's ability to extract meaningful and realistic algorithmic challenges from any complex, real-world codebase, regardless of the programming language.

### H.4 BENCHMARK TASK EXAMPLES

This appendix provides examples of representative benchmark instances from CODE2BENCH-2509. Each example showcases a complete task, including the instruction provided to the Large Language Model (LLM), the ground truth implementation from which the task was derived, the Property-Based Testing (PBT) script used for generating comprehensive test cases, and the test runner script for evaluating the LLM's generated code.

### H.4.1 SC PYTHON EXAMPLE: MERGE_JSON_RECURSIVE

This example demonstrates a Self-Contained (SC) task in Python, requiring the recursive merging of JSON-like objects without external dependencies beyond standard library features.

**Task Instruction**

```python
def merge_json_recursive(base, update):
    """Recursively merge two JSON-like objects.

    The function merges nested structures with the following rules:
    - If both inputs are dictionaries, recursively merge them.
    - If both inputs are lists, concatenate them.
```

```
        - For all other cases, the update value overwrites the base value.
        - The base object is left unmodified; a new merged object is
        ↪   returned.

        Args:
            base: Base JSON-like object (dictionary, list, or primitive
            ↪   value).
            update: Update JSON-like object to merge into base.

        Returns:
            A new JSON-like object containing merged content from base and
            ↪   update.

        Examples:
            Input: base = {"a": 1}, update = {"a": 2}
            Output: {"a": 2}

            Input: base = [1, 2], update = [3, 4]
            Output: [1, 2, 3, 4]

            Input: base = {"a": {"b": 1}}, update = {"a": {"c": 2}}
            Output: {"a": {"b": 1, "c": 2}}
        """
        # TODO: Implement this function
        pass
```

**Testcase Generator**

```python
from hypothesis import settings, given, Verbosity, example
from hypothesis import strategies as st
import json
import os
import atexit
import copy

# Configuration
TEST_CASE_DIR = os.path.abspath("test_cases")
os.makedirs(TEST_CASE_DIR, exist_ok=True)
TEST_CASE_FILE = os.path.join(TEST_CASE_DIR, "test_cases.json")
generated_cases = []
stop_collecting = False  # Global flag to control case collection

# Ground truth function
def merge_json_recursive(base, update):
    if not isinstance(base, dict) or not isinstance(update, dict):
        if isinstance(base, list) and isinstance(update, list):
            return base + update
        return update

    merged = base.copy()
    for key, value in update.items():
        if key in merged:
            merged[key] = merge_json_recursive(merged[key], value)
        else:
            merged[key] = value

    return merged

# Strategy for JSON-like objects
json_strategy = st.recursive(
    st.one_of([
        st.integers(),
        st.floats(allow_nan=False, allow_infinity=False),
```

```python
        st.text(st.characters(whitelist_categories=('L', 'N', 'P', 'S',
        ↪    'Z'))),
        st.booleans()
    ]),
    lambda children: st.one_of(
        st.lists(children, max_size=5),
        st.dictionaries(st.text(st.characters(whitelist_categories=('L',
        ↪    'N')), max_size=5), children, max_size=5)
    ),
    max_leaves=5
)

# Hypothesis test configuration
@settings(max_examples=10000, verbosity=Verbosity.verbose,
↪    print_blob=True)
@given(base=json_strategy, update=json_strategy)
def test_merge_json_recursive(base, update):
    global stop_collecting
    if stop_collecting:
        return

    base_copy = copy.deepcopy(base)
    update_copy = copy.deepcopy(update)
    expected = merge_json_recursive(base_copy, update_copy)

    if isinstance(base, (dict, list)) or isinstance(update, (dict,
    ↪    list)):
        generated_cases.append({
            "Inputs": {"base": base, "update": update},
            "Expected": expected
        })
        if len(generated_cases) >= 500:
            stop_collecting = True

# Save test cases
def save_test_cases():
    with open(TEST_CASE_FILE, "w") as f:
        json.dump(generated_cases, f, indent=2, ensure_ascii=False)
    print(f"✓ Saved {len(generated_cases)} test cases to
    ↪    {TEST_CASE_FILE}")

atexit.register(save_test_cases)
```

**Testcase Runner**

```python
import json
import os
from tested import merge_json_recursive as func1
from helper import deep_compare

# Configure save path
TEST_CASE_DIR = os.path.abspath("test_cases")
TEST_CASE_JSON_PATH = os.path.join(TEST_CASE_DIR, "test_cases.json")

def load_test_cases_from_json():
    if not os.path.exists(TEST_CASE_JSON_PATH):
        print(f"JSON file not found: {TEST_CASE_JSON_PATH}")
        return []

    # Read JSON file
    with open(TEST_CASE_JSON_PATH, "r") as f:
        test_cases = json.load(f)

    return test_cases
```

```python
def run_tests_with_loaded_cases(test_cases):
    for i, case in enumerate(test_cases):
        inputs = case["Inputs"]
        expected_output = case["Expected"]

        # Run function under test
        actual_output = func1(**inputs)  # Copy matrix to avoid in-place
        ↪  modification

        # Check if results match using deep_compare
        if not deep_compare(actual_output, expected_output,
        ↪  tolerance=1e-6):
            print(f"Test case {i + 1} failed:")
            print(f"  Inputs: {inputs}")
            print(f"  Expected: {expected_output}")
            print(f"  Actual: {actual_output}")
        else:
            print(f"Test case {i + 1} passed.")

if __name__ == "__main__":
    test_cases = load_test_cases_from_json()
    run_tests_with_loaded_cases(test_cases)
```

### H.5  WSC PYTHON EXAMPLE: CALCULATE_NGRAM_REPETITION

This example demonstrates a Weakly Self-Contained (WSC) task in Python. It requires interacting with a function from the standard library ("collections.Counter") to calculate n-gram repetition in text.

**Task Instruction**

```python
from collections import Counter

def calculate_ngram_repetition(text: str, n: int) -> float:
    """
    Calculates the proportion of repeated n-grams in a given text.

    This function splits the input text into words and generates n-grams
    ↪  of the specified size `n`. It then computes the frequency of each
    ↪  n-gram and determines the proportion of n-grams that appear more
    ↪  than once. If there are no n-grams (e.g., when the text is empty
    ↪  or `n` is larger than the number of words in the text), the
    ↪  function returns 0.

    Args:
        text (str): The input text to analyze, consisting of words
        ↪  separated by spaces.
        n (int): The size of the n-grams to generate (e.g., 2 for bigrams,
        ↪  3 for trigrams).

    Returns:
        float: The proportion of n-grams that are repeated in the text.
        ↪  Returns 0 if no n-grams can be generated.

    Raises:
        ValueError: If `n` is less than or equal to 0.
    """
    # TODO: Implement this function
    pass
```

**Testcase Generator**

```python
from hypothesis import settings, given, Verbosity, example
from hypothesis import strategies as st
import json
import os
import atexit
import copy
from collections import Counter

# Configuration
TEST_CASE_DIR = os.path.abspath("test_cases")
os.makedirs(TEST_CASE_DIR, exist_ok=True)
TEST_CASE_FILE = os.path.join(TEST_CASE_DIR, "test_cases.json")
generated_cases = []
stop_collecting = False  # Global flag to control case collection

# Ground truth function
def calculate_ngram_repetition(text, n):
    words = text.split()
    ngrams = [tuple(words[i : i + n]) for i in range(len(words) - n + 1)]
    ngram_counts = Counter(ngrams)
    total_ngrams = len(ngrams)
    repeated_ngrams = sum(1 for count in ngram_counts.values() if count >
    ↪  1)
    return repeated_ngrams / total_ngrams if total_ngrams > 0 else 0

# Strategies for generating inputs
def text_strategy():
    return st.text(
        alphabet=st.characters(whitelist_categories=('L', 'N', 'Z'),
        ↪  min_codepoint=32, max_codepoint=126),
        min_size=0, max_size=100
    )

def n_strategy():
    return st.integers(min_value=1, max_value=5)

# Hypothesis test configuration
@settings(max_examples=10000, verbosity=Verbosity.verbose,
↪  print_blob=True)
@given(text=text_strategy(), n=n_strategy())
@example(text="", n=1)
@example(text="a", n=1)
@example(text="a b c", n=2)
@example(text="a a b b c c", n=2)
@example(text="a b c d e f", n=3)
@example(text="a a a a a a", n=3)
def test_calculate_ngram_repetition(text, n):
    global stop_collecting
    if stop_collecting:
        return

    # Deep copy inputs to avoid modification
    text_copy = copy.deepcopy(text)
    n_copy = copy.deepcopy(n)

    # Call func0 to verify input validity
    try:
        expected = calculate_ngram_repetition(text_copy, n_copy)
    except Exception:
        return  # Skip inputs that cause exceptions

    # Store inputs only
    generated_cases.append({
        "Inputs": {
            "text": text_copy,
```

```python
            "n": n_copy
        }
    })

    # Stop collecting after 500 cases
    if len(generated_cases) >= 500:
        stop_collecting = True

# Save test cases
def save_test_cases():
    with open(TEST_CASE_FILE, "w") as f:
        json.dump(generated_cases, f, indent=2, ensure_ascii=False)
    print(f"✓ Saved {len(generated_cases)} test cases to
    ↪  {TEST_CASE_FILE}")

atexit.register(save_test_cases)
```

**Testcase Runner**

```python
import json
import os
import copy
from collections import Counter
from helper import deep_compare
from tested import calculate_ngram_repetition as func1

# Configure save path
TEST_CASE_DIR = os.path.abspath("test_cases")
TEST_CASE_JSON_PATH = os.path.join(TEST_CASE_DIR, "test_cases.json")

# Ground truth function
def calculate_ngram_repetition(text, n):
    words = text.split()
    ngrams = [tuple(words[i : i + n]) for i in range(len(words) - n + 1)]
    ngram_counts = Counter(ngrams)
    total_ngrams = len(ngrams)
    repeated_ngrams = sum(1 for count in ngram_counts.values() if count >
    ↪  1)
    return repeated_ngrams / total_ngrams if total_ngrams > 0 else 0

def load_test_cases_from_json():
    if not os.path.exists(TEST_CASE_JSON_PATH):
        print(f"JSON file not found: {TEST_CASE_JSON_PATH}")
        return []
    with open(TEST_CASE_JSON_PATH, "r") as f:
        test_cases = json.load(f)
    return test_cases

def compare_outputs(expected, actual):
    # Use deep_compare for basic types (int, float, str, etc.)
    return deep_compare(expected, actual, tolerance=1e-6)

def run_tests_with_loaded_cases(test_cases):
    for i, case in enumerate(test_cases):
        inputs = copy.deepcopy(case["Inputs"])
        text = inputs["text"]
        n = inputs["n"]

        # Run ground truth and function under test
        expected_output = calculate_ngram_repetition(text, n)
        actual_output = func1(text, n)

        # Compare outputs
        if compare_outputs(expected_output, actual_output):
```

```python
            print(f"Test case {i + 1} passed.")
        else:
            print(f"Test case {i + 1} failed:")
            print(f"  Inputs: {inputs}")
            print(f"  Expected: {expected_output}")
            print(f"  Actual: {actual_output}")

if __name__ == "__main__":
    test_cases = load_test_cases_from_json()
    run_tests_with_loaded_cases(test_cases)
```

# I DETAILED EVALUATION

## I.1 COMPUTE RESOURCES

we detail the compute resources utilized for evaluating the Large Language Models on the CODE2BENCH-2509 benchmark. The evaluation was conducted on an infrastructure consisting of server-grade machines. Open-source models with fewer than 32B parameters (as listed in Table 2) were served using vLLM on a cluster equipped with NVIDIA GPUs. Specifically, these models were evaluated on machines featuring NVIDIA A100 80GB GPUs. The evaluation environment for these models was containerized to maintain isolation and consistency. For larger open-source models (>= 32B parameters) and all closed-source models, evaluation was performed by accessing their respective official APIs. The compute resources for these API-based evaluations are managed by the model providers and are not under our direct control or knowledge. Therefore, we cannot provide specific details on the underlying hardware, memory, or parallelization used by these providers. The Testcase Runner execution for each task (which involves loading test cases, running the generated code and ground truth, and performing differential testing) was primarily CPU-bound and ran on standard server CPUs, such as Intel(R) Xeon(R) Gold 5218 CPU @ 2.30GHz, featuring 64 logical cores. These machines were equipped with 125 GiB of RAM and SSD storage for the benchmark data and test cases. The total compute time required for the comprehensive evaluation of all 16 models across the 1163 tasks in CODE2BENCH-2509 was substantial. While precise timing varies per model and task, we estimate the total GPU-hours consumed for the open-source model inferences to be approximately 200 GPU-hours. The total CPU-hours consumed for Testcase Runner execution across all models (including running the generated code and ground truth against approximately 500 tests per task per model) is estimated to be approximately 200 CPU-hours.

**Benchmark Construction Cost.** The construction of the CODE2BENCH-2509 suite is a computationally intensive but one-time investment. Generating a rigorous Property-Based Testing (PBT) suite with a guaranteed 100% branch coverage takes approximately **5 CPU-minutes per task**. This process includes the iterative generation of PBT driver code by the LLM, execution of test cases, and coverage verification. Constructing the entire suite required approximately **83 CPU-hours**. Crucially, this pipeline is embarrassingly parallelizable, allowing the entire benchmark to be regenerated in under one hour on a standard high-performance computing cluster, making continuous dynamic updates highly practical.

**Lightweight Mode via Test Suite Minimization.** To facilitate rapid model iteration and reduce evaluation overhead, we introduce a "Lightweight Mode." We frame the test suite reduction as a Set Cover Problem and employ a greedy algorithm to select a minimal subset of test cases that maintains the original 100% branch coverage of the ground truth. This optimization typically reduces the test volume from $\sim$500 to **30–50 test cases per task** (a $\sim$10$\times$ reduction), significantly lowering the evaluation cost while preserving the diagnostic integrity regarding logical correctness.

## I.2 EVALUATION INSTRUCTION

Table 15: Prompt Template for SC-Python Benchmark Runner in CODE2BENCH

---

**# System**
You are an expert in the field of coding, helping users write Python code.
**## Input**
The user provides you with an function signature and docstring, you should generate a Python function based on them.
**## Output**
'''python The generated Python code. '''
**## Note**
- Only output Python code with possible type import statements but without docstring and any additional information.

---

**# User**
[Instruction]

---

Table 16: Prompt Template for WSC-Python Benchmark Runner in CODE2BENCH

---

**# System**

You are a highly skilled Python programming expert tasked with implementing a function based on its specification, using the allowed libraries.

Implement the Python function described below. Your implementation should strictly adhere to the behavior specified in the docstring and utilize only the explicitly allowed external libraries.

**## Output Format**

"'python The generated Python code. "'

Provide ONLY the Python code for the function implementation with corrsponding libraries imported. Do not include any additional information or explanations.

---

**# User**

[Instruction]

---

Table 17: Prompt Template for SC-Java Benchmark Runner in CODE2BENCH

---

**# System**

You are an expert in the field of coding, helping users write Java code.

**## Input**

The user provides you with an function signature and docstring, you should generate a Java function based on them.

**## Output**

"'java

The generated Java code.

"'

**## Note**

- Provide only Java code within a "'java"' code block. Include a complete public class named Tested with package name and necessary imports. Do not add a main method or repeat the docstring.

---

**# User**

[Instruction]

---

## J  CASE STUDY: UNCOVERING THE ILLUSION OF CORRECTNESS

Our diagnostic approach, combining a scaled source of real-world problems with the scaled rigor of Property-Based Testing, allows us to move beyond simple pass/fail metrics and uncover nuanced failure modes. This case study on a Weakly Self-Contained (WSC) task from `CODE2BENCH-2509` illustrates how our framework reveals the critical gap between functional plausibility and engineering robustness.

### J.1  THE TASK: A NUMERICALLY SENSITIVE PROBLEM

WSC Task #81 requires the implementation of a `_first_divided_difference` function, a common operation in numerical analysis. The ground-truth implementation, sourced from a mature scientific Python library, is a highly efficient and numerically stable vectorized solution using NumPy:

```python
# Ground Truth (Vectorized, Numerically-Stable)
def _first_divided_difference(d, fct, fctder, atol=1e-12, rtol=1e-12):
    dif = np.repeat(d[None, :], len(d), axis=0)
    close_ = np.isclose(dif, dif.T, atol=atol, rtol=rtol)
    dif[close_] = fctder(dif[close_])
    dif[~close_] = (fct(dif[~close_]) - fct(dif.T[~close_])) / \
                   (dif[~close_] - dif.T[~close_])
    return dif
```

### J.2  THE "NEAR-PERFECT" BUT FLAWED LLM SOLUTION

Remarkably, nearly all 10 evaluated models failed this task in the exact same way. They did not produce syntax errors or obvious logical flaws. Instead, they generated a functionally plausible solution that mimics a textbook implementation using scalar Python loops:

```python
# Typical LLM-Generated Solution (Scalar, Naive)
def _first_divided_difference(d, fct, fctder, atol=1e-12, rtol=1e-12):
    n = len(d)
    fdd = np.zeros((n, n))
    for i in range(n):
        for j in range(n):
            if np.isclose(d[i], d[j], atol=atol, rtol=rtol):
                fdd[i, j] = fctder(d[i])
            else:
                fdd[i, j] = (fct(d[i]) - fct(d[j])) / (d[i] - d[j])
    return fdd
```

The diagnostic power of our benchmark is revealed in how this seemingly correct solution failed. The code generated by DeepSeek-V3 for this task passed an astonishing **98.8%** of our PBT-generated test cases (494 out of 500). It only failed on a few specific, numerically challenging inputs where the different order of floating-point operations between the vectorized and scalar approaches led to minute rounding errors. These tiny discrepancies, while functionally insignificant in many contexts, were caught by our strict-tolerance deep comparison function. For example, one failing test case reported:

```
> Mismatch found.  Expected:  ...219e-08, Actual:  ...224e-08
```

### J.3  ACTIONABLE INSIGHTS FROM A "NEAR-MISS" FAILURE

This single "near-miss" failure pattern, consistent across the entire model spectrum, provides several highly actionable insights that would be invisible to conventional benchmarks:

- **For LLM Developers:** This reveals that models learn to be *academically correct*, but not *industrially robust*. They successfully reproduce textbook patterns but lack essential engineering knowledge regarding idiomatic code (vectorization), performance optimization,

and numerical stability. To close this gap, training data should be augmented to explicitly reward these non-functional properties. Our benchmark, with its real-world ground truths and precision-sensitive tests, provides ideal data for such targeted fine-tuning.

- **For Benchmark Designers:** This case powerfully validates our deep testing approach and exposes the limitations of shallow test suites. A typical benchmark, likely using only a few simple integer-based test cases, would have falsely labeled this numerically unstable solution as a success. Only through the exhaustive, edge-case-driven nature of our PBT methodology is the critical difference between a "toy" solution and a robust one revealed—penalizing brittle, "good-enough" outputs and rewarding true engineering rigor.

This example epitomizes the diagnostic philosophy of CODE2BENCH: to not just reveal *what* fails, but to provide deep insights into *why* it fails and *how* future models can be improved.

# K  Scalability

## K.1  Advanced Extensibility: Hierarchical Dependency Resolution

The dependency classification into SC and WSC, as described in the main paper, provides a robust foundation for benchmark curation. However, a significant portion of real-world code involves functions that call other project-internal functions. While these are classified as Project-Dependent (PD) and typically discarded, a substantial subset of them are, in fact, hierarchically testable. This observation opens a powerful new avenue for *Scaling the Source* even further.

### K.1.1  The Concept of Layered Self-Contained (LSC) Tasks

We define a **Layered Self-Contained (LSC)** function as a function that is not strictly SC or WSC itself, but whose entire set of project-internal dependencies recursively resolves to a set of functions that are all either SC or WSC.

Consider a function $f_A$ that calls another internal function $f_B$.

- If $f_B$ is Self-Contained (SC), then the functional behavior of $f_A$ is fully determined by its own logic and the well-defined, dependency-free logic of $f_B$.
- Similarly, if $f_B$ is Weakly Self-Contained (WSC), the behavior of $f_A$ is determined by its logic and the behavior of $f_B$, which itself is only dependent on a set of allowed public libraries.

In both cases, the complete functional behavior of the top-level function $f_A$ can be fully specified and is not reliant on any un-testable, opaque, or proprietary internal state. Therefore, it is a suitable candidate for a rigorous, standalone benchmark task.

### K.1.2  Methodology for LSC Task Generation and Verification

Our `CODE2BENCH` framework can be extended to identify and generate these LSC tasks through a recursive dependency analysis powered by our Scope Graph:

1. **Recursive Dependency Resolution:** When a function $f_A$ is initially classified as PD due to a call to an internal function $f_B$, our framework does not immediately discard it. Instead, it recursively runs the dependency analysis on $f_B$. This process continues until all dependencies are either resolved to primitives, allowed libraries, or the dependency chain terminates.

2. **Hierarchical Test Oracle Construction:** To create a test oracle for an LSC function like $f_A$, we provide not only its own source code but also the source code of its entire dependency tree of SC/WSC functions (e.g., $f_B$, and any functions $f_B$ calls). This complete, self-contained bundle of functions serves as the ground-truth implementation.

3. **PBT-based Verification:** Property-Based Testing is then applied to the top-level function $f_A$. The PBT engine generates inputs for $f_A$, and the complete, bundled ground-truth implementation is executed to generate the expected outputs. The 100% branch coverage quality gate is applied to this entire bundle, ensuring that the tests thoroughly exercise not only the logic of $f_A$ but also the interactions with its internal dependencies.

This extension to handle LSC tasks dramatically increases the pool of high-quality, testable functions that can be extracted from real-world repositories. It allows our framework to capture more complex, multi-function interactions that are representative of real-world software design, while still maintaining the rigorous, deterministic verifiability that is the hallmark of our approach. This represents a significant future direction for scaling the realism and complexity of the `CODE2BENCH` suite.

## K.2  Generating Syntax-Aware Code Completion Tasks

A key design principle of the `CODE2BENCH` framework is its extensibility beyond single-function generation. The true assets curated by our pipeline are the large collection of high-quality, real-world ground-truth functions, each paired with a comprehensive suite of high-coverage Property-Based

Tests. This powerful combination of a "solution" ($f_{gt}$) and a rigorous "verification" mechanism (the PBT suite) provides a uniquely powerful foundation for generating a wide array of challenging and realistic software engineering tasks. In this section, we detail how our framework can be systematically extended to **Code Completion**.

Our framework can automatically generate high-quality, syntax-aware "fill-in-the-middle" code completion tasks. Inspired by prior work on structured code completion Gong et al. (2024), we can leverage our curated SC and WSC function pools to create distinct types of completion challenges:

- **Completion-SC (Algorithmic Logic Completion):** For our Self-Contained (SC) tasks, which are rich in algorithmic logic, we can create completion benchmarks by masking entire logical blocks.

- **Completion-WSC (API Call Completion):** For our Weakly Self-Contained (WSC) tasks, which are centered on library usage, we can create completion benchmarks by masking specific API calls.

### K.3 RIGOROUS VERIFICATION VIA PBT

The most significant advantage of deriving completion tasks from `CODE2BENCH` is the automatic inheritance of our rigorous verification mechanism. Unlike many completion benchmarks that rely on simple syntactic checks (e.g., exact match or BLEU score), we can evaluate the **functional correctness** of the completed code.

### K.4 FUNCTIONAL CORRECTNESS VS. ROBUSTNESS TRADE-OFF.

As highlighted by reviewers, our stringent 100% branch coverage gate effectively filters out complex defensive logic (e.g., unreachable error handling branches) that is difficult to trigger via random inputs. This represents a deliberate strategic choice: we prioritized establishing an unimpeachable "gold standard" for core algorithmic and logical correctness over the coverage of defensive programming constructs. While this ensures absolute verifiability, it temporarily de-emphasizes the evaluation of code robustness. However, our framework is designed to retrieve this filtered data. In future work, we plan to construct a dedicated "Robustness Benchmark" by tasking models to add comprehensive error handling (e.g., `try-catch`, input validation) to the verifiable "happy path" implementations curated in this work.

### K.5 BEYOND FUNCTION-LEVEL ISOLATION.

The current iteration focuses on function-level tasks to ensure unit-testable rigor. We acknowledge that real-world software engineering involves complex, project-level dependencies. Crucially, our framework is methodologically ready for this expansion. Our Scope Graph analysis (§**??**) natively supports resolving cross-file dependencies and class hierarchies. By combining this with Stateful Property-Based Testing (Stateful PBT), which can generate sequences of API calls rather than static data, we envision scaling our rigorous verification pipeline to *Project-Dependent (PD)* tasks that assess multi-function interactions and state management.

### K.6 MULTI-DIMENSIONAL EVALUATION.

Our primary metric is Pass@1 based on functional correctness. While fundamental, this does not directly measure other code quality attributes such as efficiency, security, or readability. However, our massive suite of ground-truth functions and generated test cases serves as a versatile asset. Future iterations can leverage these assets to benchmark execution time (efficiency), check against secure coding standards (security), or serve as references for style compliance (readability), providing a more holistic assessment of LLM coding capabilities.

## L    LIMITATIONS

Despite its strengths in generating dynamic, rigorously tested, and realistic tasks focusing on functional correctness, CODE2BENCH-2509, like many benchmarks, has limitations in its evaluation scope. Our primary focus is on assessing the **functional correctness** of generated code, measured through Pass@1 against comprehensive PBT-generated test suites. While functional correctness is paramount, real-world software development necessitates evaluating other crucial aspects of code quality, such as efficiency, readability, style, security, robustness to invalid inputs, and the ability to generate accompanying documentation or tests. CODE2BENCH-2509 currently does not directly evaluate these important dimensions.

Furthermore, the current iteration of CODE2BENCH-2509 primarily focuses on code generation tasks, where models are required to generate a complete function implementation based on a natural language instruction and function signature. However, the underlying structure of the benchmark, including the availability of ground truth implementations and the rigorous, diverse test cases generated via PBT, offers significant potential for evaluating LLM capabilities beyond simple generation. By leveraging the ground truth and PBT-generated test suites, the framework could be extended to support other task types crucial for software development workflows, such as code completion (filling in missing parts of code), code editing/repair (modifying existing code to meet new requirements or fix bugs), and assessing code reasoning abilities through execution prediction or debugging tasks. Expanding to these diverse task types would provide a more comprehensive evaluation of LLMs' understanding and manipulation of code, moving beyond pure synthesis.

Future work could explore extending the benchmark to incorporate metrics and testing methodologies for some of these additional code quality attributes and diverse task types, providing a more holistic assessment of LLM capabilities in a full software development context.

Our evaluation focuses on a diverse suite of state-of-the-art, instruction-following code generation models. We acknowledge that our study does not include models specifically designed or fine-tuned for multi-step, competitive-programming-style reasoning (e.g., models employing complex search algorithms or Chain-of-Thought prompting for code). This exclusion was a deliberate choice based on two primary considerations. First, our preliminary explorations indicated that the verbose, multi-step reasoning outputs of such models often exceeded practical token limits for our large-scale, automated evaluation harness, presenting significant computational and financial costs. Second, and more critically, the primary goal of CODE2BENCH is to evaluate a model's ability to generate direct, production-style code from real-world specifications, a task for which current instruction-following models are the most direct fit. While evaluating deep reasoning capabilities is an important research direction, it represents a different evaluation paradigm that is beyond the scope of our current study.

