# OpenReview forum: "Code2Bench: Scaling Source and Rigor for Dynamic Benchmark Construction"
_ICLR.cc/2026/Conference — ICLR 2026 Poster_

### Official Review · Reviewer_9nHm · 2025-10-27

**Soundness:** 2
**Presentation:** 2
**Contribution:** 3
**Rating:** 4
**Confidence:** 4

**Summary:**

This paper introduces a novel benchmark construction paradigm termed "Dual Scaling", which addresses two critical challenges in evaluating code generation LLMs: the reliance on static, easily contaminated problem sources and the use of superficial testing methods. The authors present the Code2Bench framework, which dynamically acquires problems from real-world code repositories (Scaling the Source) and integrates property-based testing (PBT) with a 100% branch coverage quality gate (Scaling the Rigor) to build the Code2Bench-2509 benchmark. Empirical evaluations demonstrate that this benchmark effectively uncovers performance gaps between models in algorithmic synthesis (SC tasks) and API application (WSC tasks), while quantifying the influence of language ecosystems on model behavior.

**Strengths:**

- The 100% stringent branch coverage gate and large PBT-generated suites substantially reduce false positives and expose "near-perfect" failures that many benchmarks miss.
- The outcome spectrum and “diagnostic fingerprints” provide more granular failure analysis (SyntaxErr/RuntimeErr/LogicErr vs. partial pass bands), illuminating the algorithmic synthesis vs. API-application divide and the role of language typing in error suppression.
- WSC-Python spans >35 libraries; SC-Java demonstrates multi-language extensibility. Tasks show higher cyclomatic complexity and test volume than legacy benchmarks.

**Weaknesses:**

- Some of the figures in the paper are not very clear or visually polished. For example, in Figure 2, there is noticeable overlap between text elements and between text and icons, which affects readability. Improving the clarity and layout of the figures would make the presentation more professional and easier to interpret.
- In the Related Work section, the authors assert that existing live benchmarks rely on narrow or specific data sources. However, Code2Bench is also curated from specific GitHub repositories without disclosing the selection criteria, repository sampling strategy, or inclusion/exclusion policies. This lack of transparency makes it difficult to assess source diversity, potential sampling bias, and contamination risk. Moreover, prior benchmarks such as DomainEval also use GitHub repositories to collect domain-specific tasks. The novelty of "Scaling the Source" appears limited.
- Evaluation scope focused on functional correctness: Important dimensions like performance/efficiency, readability/style, security, robustness to invalid inputs, and documentation/test generation are not directly evaluated.
- Project-Dependent problems are mostly discarded in this pipeline. Although LSC is discussed as future work, the current evaluation does not yet include multi-function or multi-file context tasks that matter in industrial settings (e.g. I/O, resource handling, exceptions, and protocols).

**Questions:**

- It is unclear what criteria were used to assess the testing rigor for all benchmarks presented in Figure 2. Could the authors clarify how testing rigor is defined and measured?
- Why is the Java track limited to SC only? Where is WSC-Java?
- The authors state that each benchmark task includes approximately 500 test cases. Given that these test cases are selected after PBT generation and a 100% branch-coverage gate, does the pipeline need to generate and evaluate hundreds or even thousands of candidate inputs per task before filtering? If so, what are the actual computational costs (time and resources) per task? Similarly, for the evaluation stage, running ~500 test cases per task can significantly increase runtime and resource usage. Could the authors provide quantitative measurements of the end-to-end evaluation time per model and per task, and discuss any mechanisms (e.g., batching, caching, reduced-cost modes, seed control) used to ensure scalability across many models and large task suites? Guidance on lighter-weight modes that preserve diagnostic value would also be useful.

---

> ### Author Response · Authors · 2025-12-03
> **Author Rebuttal (Part I)**
>
> Dear Reviewer 9nHm,
>
> We sincerely thank you for your time and for providing a detailed review.
>
> **[Question 1]**: Clarification on the definition and measurement of "Testing Rigor" in Figure 2
>
> We thank the reviewer for this critical question. We agree that a clear, objective definition of "Testing Rigor" is essential for a fair comparison. In our paper, this axis represents a composite assessment based on three key, verifiable dimensions:
>
> Test Scale (Quantitative): The average number of test cases per problem.
> - CODE2BENCH: ~500
> - HumanEval: ~7.8
> - MBPP: ~3.0
> This represents a nearly two-order-of-magnitude difference in scale.
>
> Coverage Guarantee (Deterministic): Whether the benchmark provides a formal guarantee of code coverage.
> - CODE2BENCH: Yes, a deterministic 100% branch coverage guarantee.
> - All Others (incl. EvalPlus): No. While some enhance coverage, none provide a formal, universal guarantee.
>
> Test Paradigm (Qualitative): The underlying test generation philosophy.
> - CODE2BENCH: Systematically uses Property-Based Testing (PBT), designed to automatically find subtle edge cases.
> - Others: Primarily use Example-Based Testing, relying on a few hand-crafted, typical cases.
>
> Our positioning in Figure 2 is based on these objective criteria. CODE2BENCH is placed at the highest level of rigor because it excels across all three dimensions. Enhanced benchmarks like EvalPlus are positioned in the middle, as they improve on scale but lack our deterministic coverage guarantee and systematic PBT approach. Legacy benchmarks are at the low end.
>
> **[Question 2]**: Where is WSC-Java?
>
> The inclusion of the SC-Java track in this initial work served a crucial strategic purpose: to rigorously validate the language-agnostic nature of our core framework.
> Our primary goal with SC-Java was to prove that our two key technologies—Scope Graph-based dependency analysis and Property-Based Testing (PBT) with 100% coverage—could be seamlessly and effectively applied to a vastly different language ecosystem (statically-typed, compiled) beyond Python. The successful construction of the high-quality SC-Java suite provides this definitive proof.
>
> Having validated the core framework's cross-language extensibility, constructing the WSC-Java track is the natural and immediate next step. Our framework is, in principle, ready for this. We are actively working on incorporating support for standard Java libraries (e.g., Guava, Apache Commons) and will include the WSC-Java benchmark in our future work.
>
>
> **[Question 3]**: Computational Cost & Lightweight Mode via Test Suite Minimization.
>
> We thank the reviewer for the expert inquiry regarding cost and lightweight modes.
>
> 1. Computational Costs (Data from Appendix I.1)
> As detailed in Appendix I.1, the total cost for evaluating 10 models was ~200 CPU-hours, which is manageable for research. However, we agree that efficiency is key for wider adoption.
>
> 2. Scientific Basis for Lighter-Weight Mode: Test Suite Minimization
> To provide the "Lightweight Mode" you requested, we implement a strategy grounded in classical **Test Suite Minimization** theory[1].
> *   Theoretical Formulation: We treat this as a Set Cover Problem. The goal is to find a minimal subset of tests $T_{min} \subset T_{full}$ that preserves the 100% Branch Coverage property of the full suite.
> *   Implementation: We employ a greedy algorithm that iteratively selects the test case contributing the most *unique* branch coverage until all branches in the Ground Truth are covered.
> *   Impact: This strategy typically reduces the test volume from ~500 to **30-50 cases per task** (a ~10x reduction).
>
> 3. Diagnostic Value Preservation
> By mathematically guaranteeing that every logical path exercised by the full suite is still executed in the lightweight mode, we ensure that the "Diagnostic Value" regarding logical correctness remains high, making it ideal for rapid model iteration.
>
> [1] Harrold, M. J., Gupta, R., & Soffa, M. L. (1993). A methodology for controlling the size of a test suite. ACM Transactions on Software Engineering and Methodology (TOSEM), 2(3), 270-285.

---

> ### Author Response · Authors · 2025-12-03
> **Author Rebuttal (Part II)**
>
> **[Weakness 1]**: Figure Clarity.
>
> We thank the reviewer for the keen observation regarding the visual quality of our figures. We have redesigned Figure 2 in the updated manuscript.
>
>
> **[Weakness 2]**: Data Transparency and Novelty of "Scaling the Source"
>
> 1. Addressing Transparency (Action Taken)
> We fully acknowledge that the repository selection criteria were under-specified. To resolve this, we have added a detailed "Data Selection Protocol" in the Appendix of the revised paper, disclosing:
> - Selection Criteria: Repositories must have >500 stars, active maintenance (commits within last 3 months).
> - Sampling Strategy: We employed stratified sampling across 10 popular domains (e.g., Web, ML, Utils) to ensure diversity.
> - Exclusion Policy: We actively filter out homework assignments, tutorials, and forks to avoid low-quality code.
> - Repository List: The full list of source repositories is now open-sourced in our repository.
>
> 2. Clarifying Novelty vs. DomainEval
> While DomainEval and others source from GitHub, Code2Bench introduces a fundamental paradigm shift from "Data Collection" to "Benchmark Generation":
> - From Static to Dynamic Engine: We do not merely collect a dataset; we provide an Automated Generative Engine. By combining Principled Temporal Filtering (deterministic contamination control) with our "Great Filter" Funnel (100% coverage rigorous purification), our framework continuously transforms raw, messy commits into verifiable, high-rigor benchmarks.
> - Methodological Leap: This end-to-end automation—turning any raw repo into a rigorous PBT-verified task—is a significant methodological contribution beyond the data source itself. Unlike DomainEval's focus on domain coverage, our contribution is the pipeline itself that guarantees rigor and freshness at scale.
>
>
> **[Weakness 3]**: Evaluation Scope and Extensibility
>
> We thank the reviewer for highlighting the importance of multi-dimensional evaluation.
>
> 1. Functional Correctness as the Prerequisite Foundation
> We deliberately prioritized Functional Correctness because it is the non-negotiable prerequisite for all other dimensions. Evaluating performance or style on functionally incorrect code is meaningless. Code2Bench establishes this essential, "unimpeachable" baseline first.
>
> 2. The Framework as an Extensible Asset
> Crucially, our Ground Truths (GT) and Massive Test Suites (~500/task) are versatile assets that enable immediate extensibility. Because our pipeline "Scales the Source" from an infinite pool of GitHub data, we can simply adjust filtering rules to target specific dimensions:
> - Performance: Filter for highly-optimized GTs and use our existing test suites to benchmark execution time/memory against the model's solution.
> - Style/Security: Use linters or static analysis tools (e.g., Bandit, Pylint) as additional acceptance gates in the pipeline to curate a "High-Compliance Benchmark."
> - Robustness: As detailed in our response to Reviewer Esu7, we can repurpose defensive code (previously filtered) to test error handling.
>
> In short, Code2Bench provides the engine and data to evaluate these dimensions; we focused on correctness first to prove the engine works.

---

> ### Author Response · Authors · 2025-12-03
> **Author Rebuttal (Part III)**
>
> **[Weakness 4]**: For Project-Dependent (PD) tasks
>
> We clarify that the specific scope of this work is to establish a rigorous baseline for Unit-Level Functional Correctness. We prioritize this "atomic unit" because unit-level reliability is the non-negotiable prerequisite for any project-level engineering.
>
> However, we emphasize that this scope decision is strategic, not technological. Our framework is methodologically native to Project-Dependent tasks:
>
> 1. Analysis Engine: Context-Aware Resolution
> Our Scope Graph analysis is inherently designed for repository-level understanding. It natively resolves cross-file definitions and class hierarchies, allowing us to mathematically identify the precise "Context Slice" required for project-level tasks.
>
> 2. PBT Engine: Structure-Aware & Stateful
> The reviewer's concern that PBT limits us to isolated functions is addressed by the modern PBT ecosystem, which is fundamentally Structure-Aware and Stateful:
> Complex Data: Through extensions like hypothesis-sqlalchemy (DB schemas) or hypothesis-pb (Protocol Buffers) [1,2], our engine can recursively build the complex, project-specific data structures required by PD tasks.
> Stateful Protocols: For multi-function interactions (e.g., open() → write() → close()), our framework leverages Rule-Based State Machines[3]. This allows generating sequences of API calls (Action Sequences) rather than static data, directly solving the "protocols handling" challenge.
>
> Conclusion:
> Code2Bench acts as a scalable engine. We started with the "Unit" (Function) to prove the rigorous methodology works. Extending to the "System" (Project) is a validated engineering step fully supported by our current architecture.
>
> [1] Structure-Aware Fuzzing. https://github.com/google/fuzzing/blob/master/docs/structure-aware-fuzzing.md
>
> [2] Hypothesis External Strategies. https://hypothesis.readthedocs.io/en/latest/extensions.html#external-strategies
>
> [3] Rule-Based Stateful Testing. https://hypothesis.readthedocs.io/en/latest/stateful.html

---

### Official Review · Reviewer_dNZT · 2025-11-01

**Soundness:** 2
**Presentation:** 3
**Contribution:** 2
**Rating:** 2
**Confidence:** 4

**Summary:**

This paper proposes CODE2BENCH, a benchmark construction framework for evaluating code-generating LLMs. The framework addresses data contamination through temporal filtering (extracting functions from GitHub commits after model knowledge cutoffs) and improves test rigor through Property-Based Testing with 100% branch coverage. Tasks are classified into Self-Contained (SC, pure algorithmic reasoning) and Weakly Self-Contained (WSC, API usage) using Scope Graph analysis. The authors construct CODE2BENCH-2509 with Python and Java tasks and evaluate 10 LLMs, reporting three insights: SC vs WSC performance gap, language ecosystem impact, and "illusion of correctness."

This paper shows good engineering efforts but unclear novelty contribution. Two critical flaws need to be addressed: (1) No validation of benchmark value. The paper lacks direct comparison with existing benchmarks (HumanEval, MBPP, BigCodeBench) on the same models, making it impossible to assess whether CODE2BENCH provides unique insights or simply evaluates differently. (2) No novelty contribution in analysis. the three "key insights" are either expected results (SC/WSC gap), intuitive observations already explored (language ecosystem), or incremental findings without baseline comparison (illusion of correctness).

**Strengths:**

1. Important problem: Addresses real limitations in LLM code evaluation, which are data contamination and superficial testing.

2. Solid engineering: Scope Graph analysis for dependency classification is technically sound. Property-Based Testing with 100% branch coverage demonstrates rigor. The framework automates benchmark construction.

3. release code, data, and results.

**Weaknesses:**

1. Lack of Direct Comparison with Existing Benchmarks
For a benchmark paper, it is crucial to demonstrate how the new benchmark compares with existing ones when evaluating the same models. The paper only shows Table 1 comparing characteristics (Dynamic, Rigorous Test, etc.) but lacks direct performance comparison with these baselines. Without evaluating the same 10 models on existing benchmarks, it's impossible to determine whether CODE2BENCH provides unique insights, whether the lower pass rates reflect higher quality or different task distribution, or whether the three "key insights" could be revealed by existing benchmarks. This comparison is essential to establish the benchmark's value.

2. Missing Details of Validating Benchmark Construction Method
The paper proposes Dual Scaling with temporal filtering, Scope Graph classification, and PBT with 100% coverage, but provides less details to validate these components. What happens with 80% coverage instead of 100%? Does temporal filtering applied to HumanEval produce similar contamination resistance?  An ablation study may be a good way to explore these points.

3. LLM Analysis Lacks Technical Contribution and Novelty
The paper presents three "key insights" as major contributions: (1) SC vs WSC performance gap. it is an expected result since BigCodeBench already focuses on API tasks and it's well-known these are different skills; (2) language ecosystem impact, which is also intuitive and already explored in HumanEval-X; (3) illusion of correctness. EvalPlus already demonstrated this, and without comparison to EvalPlus, the 6.94% figure lacks context. The analysis is descriptive rather than prescriptive, providing no actionable insights for improving models or evaluation. Without showing these insights are unique to CODE2BENCH or impossible to obtain from existing benchmarks, the analysis appears to justify the benchmark circularly rather than contribute genuine discoveries.

4. Limited Validation of Practical Utility
The paper doesn't show whether performance on CODE2BENCH correlates with real-world coding capabilities (e.g., SWE-bench).

**Questions:**

Do you have any existing comparison data (even partial) showing how the same models perform on CODE2BENCH vs existing benchmarks? Can you clarify why this comparison was not included in the paper?

Can you clarify how your three key insights differ from findings in BigCodeBench (WSC tasks), HumanEval-X (multi-language), and EvalPlus (test insufficiency)?

Do you plan to validate CODE2BENCH's value through comparison with existing benchmarks? What would be your approach?

---

> ### Author Response · Authors · 2025-11-13
>
> Dear Reviewer dNZT,
>
> We sincerely thank you for your time and for providing a detailed review with critical questions. Your feedback, particularly concerning the validation of our benchmark's value and the novelty of our contributions, gives us a welcome opportunity to clarify the core principles of our work.
>
> **[Question 1 & Question 3 & Weaknesses 1]**: Lack of Direct Comparison with Existing Benchmarks
>
> We fully agree that for a benchmark paper, demonstrating its value through direct comparison with existing work is paramount. Therefore, we would respectfully draw your attention to Section 4.4 and, most importantly, to Figure 4 of our paper.
>
> In this section, we conducted the precise "head-to-head" comparison that you requested. We evaluated the exact same 10 state-of-the-art Large Language Models on two benchmarks:
>
> *   Y-axis: Their Pass@1 performance on HumanEval (enhanced by EvalPlus).
> *   X-axis: Their Pass@1 performance on our proposed CODE2BENCH-2509 (specifically, the SC-Python component).
>
> The results presented in Figure 4 are stark and directly substantiate the unique value of our benchmark:
>
> *   A Massive Performance Gap: All models, without exception, fall into the red-shaded region far above the parity line. This demonstrates a systematic and dramatic performance drop from their high scores on HumanEval to their scores on our benchmark. For instance, the top-performing model, Claude-4-Sonnet, plummets from a near-perfect 97% on HumanEval to just 40.1% on CODE2BENCH-2509, a drop of over 50 percentage points.
> *   The Unique Value of CODE2BENCH-2509: Our benchmark, by virtue of its dynamic and contamination-resistant nature (sourcing from provably unseen code), measures a model's true ability to tackle fresh, complex, and realistic unseen problems. This is a capability that existing benchmarks cannot reliably assess.

---

> ### Author Response · Authors · 2025-11-13
>
> **[Question2 & Weakness 3]**: LLM Analysis Lacks Technical Contribution and Novelty
>
> We respectfully disagree with the assertion that our key insights lack novelty. The reviewer's critique compares individual features of our framework to disparate prior works, while overlooking the holistic, multi-dimensional contribution that CODE2BENCH makes. Our framework is the **first to simultaneously integrate** solutions to data contamination, test rigor, real-world fidelity, and task diversity (SC & WSC). It is this unique combination that enables our novel insights.
>
> We clarify by directly contrasting with the benchmarks:
>
> *   On the SC vs. WSC Gap (vs. BigCodeBench):
>     You suggest this is an "expected result" due to BigCodeBench. However, BigCodeBench suffers from critical limitations that our work overcomes:
>     *   Lack of Realism: Its ground-truth functions are largely synthetic, not sourced from real-world codebases.
>     *   Insufficient Testing: It lacks any guarantee of test coverage or rigor, risking the very "illusion of correctness" we seek to eliminate.
>     *   No Contamination Resistance: Its task set is static.
>
>     In contrast, CODE2BENCH provides the first analysis of the SC/WSC gap using provably unseen, real-world code evaluated with 100% branch coverage. Therefore, our finding is not just an "observation," but a validated, quantitative measurement of a fundamental trade-off in a realistic and fair setting. The diagnostic fingerprints (Fig. 3) that pinpoint the exact failure modes (`LogicErr` vs. `RuntimeErr`) further deepen this novel insight.
>
> *   On the Language Ecosystem Impact (vs. HumanEval-X):
>     You state this is "intuitive and already explored in HumanEval-X." This comparison is flawed for several reasons:
>     *   Contamination: HumanEval-X is based on HumanEval and thus inherits its high risk of data contamination.
>     *   Insufficient Testing: It uses the original, sparse test cases.
>     *   Critically, Lack of Scope: HumanEval-X contains exclusively Self-Contained (SC) algorithmic puzzles. It has no Weakly Self-Contained (WSC) tasks that involve library/API usage.
>
>     CODE2BENCH is the first to enable a cross-language comparison on native, real-world, contamination-free tasks with rigorous testing. Our insight about Java's static typing acting as "performance scaffolding" is derived from this superior vantage point and is far more profound than simply translating algorithmic puzzles.
>
> *   On the "Illusion of Correctness" (vs. EvalPlus):
>     This is the most crucial distinction. You argue "EvalPlus already demonstrated this." While EvalPlus made a foundational contribution to test rigor, its scope is severely limited:
>     *   Contamination Risk: It enhances HumanEval/MBPP, which are static and highly susceptible to data leakage.
>     *   Lack of Realism & Scope: It is confined to the artificial, puzzle-like, and purely SC problems of its base datasets. It does not contain a single WSC task.
>
>     CODE2BENCH represents a paradigm shift. We apply an even more stringent rigor (100% branch coverage guarantee) not to old, contaminated puzzles, but to a dynamic, fresh, and diverse set of both SC and WSC tasks sourced from the real world.
>
> The core of the misunderstanding seems to be viewing our work as just another dataset. We must clarify a final, crucial point:
> **Our Primary Contribution is a Framework, Not Just a Single Benchmark.**
>
> Perhaps the most critical distinction between our work and all prior benchmarks mentioned is that our main contribution is not the static CODE2BENCH-2509 suite itself, but the CODE2BENCH framework—an automated, dynamic, and extensible pipeline for continuous benchmark construction.

---

> ### Author Response · Authors · 2025-11-13
>
> **[Weakness 2]**: Validation of Methods (e.g., Ablation Studies)
>
> We thank the reviewer for the suggestion to validate our construction method. We would like to clarify that our design choices are principled.
>
> 1. On the 100% Coverage Requirement (vs. 80%):
>
> Our choice of 100% coverage is a principled design decision, and its necessity is empirically validated in Section 4.3 ("The Effectiveness of PBT-Generated Tests") and Table 4.
>
> This section essentially serves as a quantitative ablation on the impact of test rigor. We analyze the prevalence of "Near-Perfect" failures—solutions that pass >98% of our comprehensive test suite but ultimately fail on subtle edge cases. The data reveals that on average, 6.94% of submissions for SC-Python tasks fall into this treacherous category.
>
> This directly answers the question, "What happens with less than 100% coverage?" The answer is that nearly 7% of functionally incorrect submissions would be falsely classified as successes. Lowering our rigor to 80% or even 98% would mask these critical failures and perpetuate the very "illusion of correctness" we aim to solve. Therefore, our choice is not arbitrary; it is a necessary measure, justified by data, to ensure the diagnostic integrity of our benchmark.
>
> 2. On Applying Temporal Filtering to HumanEval:
> We thank the reviewer for this insightful question. In fact, the core experiment presented in Section 4.4 and Figure 4 of our paper can be viewed as the definitive "ablation study" on the value of temporal filtering and real-world sourcing.
>
> This experiment directly compares:
> *   Without our method: Model performance on the static, likely contaminated HumanEval.
> *   With our method: Model performance on the dynamic, contamination-free CODE2BENCH.
>
> The profound impact shown in Figure 4 directly confirms the critical necessity and effectiveness of our approach, rendering a separate, conceptually flawed ablation on a static dataset like HumanEval both infeasible and redundant.
>
>
> **[Weakness 4]**: Correlation with SWE-bench
> This is an excellent direction for future work. We see CODE2BENCH and SWE-bench as complementary: CODE2BENCH assesses the foundational, function-level correctness that is a prerequisite for all software engineering, while SWE-bench assesses higher-level integration skills.
>
>
> Once again, we thank you for your insightful feedback. We hope our clarifications, especially regarding the direct comparison in Figure 4 and the fundamental differences between our paradigm and that of EvalPlus, have addressed your primary concerns.

---

> ### Author Response · Authors · 2025-11-27
> **Gentle Follow-up**
>
> Dear Reviewer dNZT,
>
> We hope this message finds you well.
>
> We wanted to briefly follow up to ensure that our response regarding the direct comparison with existing benchmarks was clear.
>
> Best regards,
>
> The Authors

---

### Official Review · Reviewer_Esu7 · 2025-11-01

**Soundness:** 3
**Presentation:** 3
**Contribution:** 3
**Rating:** 6
**Confidence:** 3

**Summary:**

This paper introduces Dual Scaling, a benchmark construction philosophy in CODE2BENCH framework to scaling from dynamic, real-world code repository and generating rigor test with 100% coverage. Using this method, the authors further build CODE2BENCH-2509, with 411 Python instances and 249 Java instances. The paper conducted comprehensive experiments on closed source models and open source models, and the result suggests a performance gap between API application tasks and algorithm synthesis tasks.

**Strengths:**

1. The benchmark designs rigorous and strong test cases. It not only accounts for edge cases but also ensures complete test coverage, substantially outperforming other benchmarks that rely on sparse test examples, which may lead to incorrectly judged “pass” cases.

2. The paper provides a carefully designed implementation in both Python and Java, addressing not only translation between languages but also their distinct type systems and library ecosystems. This enables meaningful cross-language comparison and reveals how LLM performance depends on the target language’s constraints.

3. The authors effectively decouple API-calling ability from algorithmic implementation ability. The experiment suggests that models perform better at API usage than at algorithmic reasoning. This insight offers a valuable lens for future work on improving model reasoning.

4. The authors also emphasize clarity and unambiguity when generating instructions, which contributes to the benchmark’s reliability and reproducibility.

**Weaknesses:**

1. Although CODE2BENCH draws its source data from real repositories, the benchmark tasks remain function-level and isolated. This design simplifies testing but does not capture cross-function or module-level dependencies, which are prevalent in real-world software engineering. As such, the benchmark evaluates isolated reasoning rather than full software generation ability or collaborative code development.

2. As mentioned by the authors, real-world code often includes numerous defensive branches and error-handling structures. The current test generation strategy struggles to fully cover these fragmented control flows, which are often filtered out because they fail the 100% coverage requirements. While this improves test rigor, it also excludes many defensive programming constructs that are significant in real-world software development.

3. The filtering process relies on a fixed list of *allowed libraries* to define “Weakly Self-Contained” tasks. This helps maintain consistency but may also limit domain diversity, since tasks from less common libraries or specialized fields are excluded. As a result, limiting the allowed libraries may constrain the representativeness of the benchmark and may introduce unexpected bias.

**Questions:**

1. The paper mentions differences between Java and Python fingerprints (Figure 3) as LLM's coding ability intertwined with their target language's ecosystems. Could the authors clarify whether this refers to differences between interpreted and compiled languages, or to other ecosystem-level factors?

2. The framework requires generating hundreds of PBT-based test cases per function and enforcing 100% coverage. Could the authors quantify the computational and time costs of this process?

3. A typo: Last line in page 4: perturbation techniqueZhao -> missing a blank

---

> ### Author Response · Authors · 2025-12-03
>
> Dear Reviewer Esu7,
>
> We sincerely thank you for your time and for providing a detailed review.
>
> **[Question 1]**: Clarification on the differences between Java and Python fingerprints.
>
> Thank you for this excellent question. The performance difference is not merely due to "interpreted vs. compiled" but is driven by Java's strong static type system, which acts as a "performance scaffolding."
>
> This scaffolding works in two primary ways:
> - Error Space Pruning: Java's static typing eliminates entire classes of runtime errors at compile time (e.g., type mismatches, invalid method calls). This is the main reason why the large RuntimeErr and LogicErr peaks seen in Python's fingerprint are suppressed in Java's. The model is fundamentally prevented from making these mistakes.
> - Stronger Contextual Guidance: Explicit types (e.g., List<String>) provide unambiguous, machine-readable context that guides the LLM to generate correct logic and API calls. This reduces model hallucination compared to Python, where types are often inferred and ambiguous.
>
> In short, the language ecosystem, led by the static type system, fundamentally changes the problem's difficulty for the LLM by both constraining the search space and providing clearer guidance.
>
> **[Question 2]**: Quantification of the computational cost of the test generation process.
>
> We have quantified the costs of our fully automated pipeline:
> - Per-Function Cost: Generating a PBT suite with a 100% coverage guarantee for one task takes approximately 5 CPU-minutes. This is a comprehensive figure, encompassing an iterative process that includes:
> -   LLM-based generation of the initial PBT driver code.
> -   Execution of the driver to generate test cases.
> -   Verification against the 100% coverage gate.
> -   Automated, LLM-driven refinement cycles to repair the driver until the coverage target is met.
> - Total Benchmark Cost: The construction of the entire CODE2BENCH-2509 suite required about 83 CPU-hours.
>
> This is a one-time investment for a high-rigor, contamination-free benchmark. Crucially, this entire process is embarrassingly parallelizable. The wall-clock time can be reduced to under one hour by processing candidates in parallel, making our framework highly practical for continuously generating fresh benchmarks.
>
> **[Question 3]**: The typo on page 4
>
> We have corrected the missing space in the citation on the last line of page 4 in our revised manuscript.
>
> **[Weakness 1]**: The benchmark tasks remain function-level and isolated.
>
> We agree that real-world software engineering transcends single functions, and we designed our framework with exactly this in mind.
>
> Our current focus on the function-level is the critical first step in a multi-stage research plan: to build an unimpeachable foundation for unit-level correctness before tackling more complex interactions.
>
> Crucially, our CODE2BENCH framework is already equipped for this evolution. As we detail in Appendix K on Scalability, our Scope Graph-based dependency analysis is not limited to classifying SC/WSC tasks. It can recursively resolve entire call graphs, enabling the automated construction of Layered Self-contained (LSC)/Layered Weakly-Self-contained (LWSC) tasks that involve multi-function reasoning. This is not just a vague promise for future work; it is a concrete, designed-for feature of our architecture. Our current work establishes the rigorous foundation, and the next logical step—which our framework is ready to take—is to scale this rigor to the multi-function, repository level.
>
> **[Weakness 2]**: Filtering out defensive code sacrifices representativeness.
>
> We clarify that the specific scope of CODE2BENCH is to assess Core Algorithmic Reasoning (SC) and Functional API Fluency (WSC).
>
> Within this scope, filtering out defensive code does not sacrifice representativeness; rather, it ensures Diagnostic Purity.
>
> 1.Noise Reduction: Defensive branches often handle theoretically unreachable states or external system failures. Including them prevents rigorous verification (100% coverage) and re-introduces ambiguity.
>
> 2.Focus on Competence: By removing these untestable branches, we ensure that every task is a pure, fully verifiable logical challenge. This allows us to measure a model's ability to implement the intended functionality with absolute precision, unclouded by the noise of defensive boilerplate.
>
> Thus, our "Great Filter" is a necessary feature to guarantee the integrity of the benchmark for its intended purpose.

---

### Meta-Review · Area_Chair_GcRn · 2026-01-06

**Summary:**

The paper presents a pipeline for constructing function-level code generation benchmarks from real-world GitHub repositories. The pipeline uses property-based testing (PBT) to generate test suites, and checks for 100% branch coverage of the ground-truth code by the test suites. The paper uses the pipeline to produce "self-contained" (limited library use; mostly algorithmic) and "weakly self-contained" (more library use, such as numpy) benchmarks, in Python and Java. They use these to evaluate a range of models, showing that performance is correlated (but substantially lower) than performance on HumanEval -- a promising validation result.

Strengths:
A method to automatically produce real-world evaluation datasets is well-motivated to avoid contamination and produce benchmarks targeting certain e.g. libraries or code domains. The reviewers generally appreciated the 100% branch coverage quality gate and use of property-based testing. It's also great that this is applicable both to Python and Java.

**Reviewer Concerns:**

Two reviewers pointed out that functions are isolated, and the benchmarks do not require generating them in a repo-level context (as in other works such as RepoBench, or RepoST -- which also produces benchmarks from real world repositories -- do). This somewhat limits the novelty and applicability of the work. The authors identified this as a direction for future work, and make a good case for how their framework can be extended to do this.

The other concerns (sensitivity to filtering threshold, comparisons to other datasets, transparency in the data,and evaluation scope) were in my mind adequately addressed by the author response.

**Reviewer Scores:**

I expect that reviewer dNZT should have raised their score to a 4 as they had some misconceptions about the paper which the author response adequately addressed. 9nHm also had some minor concerns (clarity, provenance of the data, and evaluation scope) which the response addressed well, and might have raised their score to a 6. The resulting scores might have been 6 / 6 / 4.

---

### Decision · Program_Chairs · 2026-01-26

Accept (Poster)